# A biochemically-interpretable machine learning classifier for microbial GWAS

Erol S. Kavvas [1], Laurence Yang[2], Jonathan M. Monk [1], David Heckmann [1] & Bernhard O. Palsson [1,3✉]

Current machine learning classifiers have successfully been applied to whole-genome sequencing data to identify genetic determinants of antimicrobial resistance (AMR), but they lack causal interpretation. Here we present a metabolic model-based machine learning classifier, named Metabolic Allele Classifier (MAC), that uses flux balance analysis to estimate the biochemical effects of alleles. We apply the MAC to a dataset of 1595 drug-tested *Mycobacterium tuberculosis* strains and show that MACs predict AMR phenotypes with accuracy on par with mechanism-agnostic machine learning models (isoniazid AUC = 0.93) while enabling a biochemical interpretation of the genotype-phenotype map. Interpretation of MACs for three antibiotics (pyrazinamide, para-aminosalicylic acid, and isoniazid) recapitulates known AMR mechanisms and suggest a biochemical basis for how the identified alleles cause AMR. Extending flux balance analysis to identify accurate sequence classifiers thus contributes mechanistic insights to GWAS, a field thus far dominated by mechanism-agnostic results.

[1] Department of Bioengineering, University of California, San Diego, CA, USA. [2] Department of Chemical Engineering, Queen's University, Kingston, ON K7L 3N6, Canada. [3] Department of Pediatrics, University of California, San Diego, CA, USA. ✉email: palsson@ucsd.edu

**M**ycobacterium tuberculosis (TB) claims 1.6 million lives annually and resists eradication through evolution of antimicrobial resistance (AMR)[1]. To elucidate AMR mechanisms, researchers have applied machine learning approaches to large-scale genome sequencing and drug-testing datasets for identifying genetic determinants of AMR[2–7]. While current machine learning approaches have provided a predictive tool for microbial genome-wide association studies (GWAS), such black-box models are incapable of mechanistically interpreting genetic associations. Such a limitation has become increasingly apparent in TB, where numerous experimental studies have shown that AMR-associated genetic variants often reflect network-level metabolic adaptations to antibiotic-induced selection pressures (Supplementary Fig. 1, Supplementary Table 1)[8–12]. These studies show that identified genetic associations have corresponding network-level associations that are highly informative of AMR mechanisms. However, current GWAS results only provide predictions for which alleles are most important, not their functional effects. Therefore, machine learning models that incorporate biochemical network structure may naturally extend GWAS results by estimating functional effects of identified alleles, leading to an enhanced understanding of AMR[13–15].

Over the past couple of decades, the computational analysis of biochemical networks in microorganisms has been advanced through the use of genome-scale models (GEMs)[16,17]. By computing metabolic flux states (see Glossary for definition of terms) consistent with imposed biological constraints, GEMs have been shown to predict a range of cellular functions, making them a valuable tool for analyzing multi-omics datasets[18]. Although GEMs are transparent genotype-phenotype models, they are largely outperformed by machine learning models in direct comparisons of prediction accuracy. Approaches have thus been developed that integrate meaningful GEM computations with predictive black-box machine learning to enable white-box interpretations of data[19]. These approaches have worked well for endogenous metabolomics data by using the GEM to directly transform the measurements to meaningful inputs for black box machine learning.

This approach, however, may not be amenable to analyzing microbial GWAS data, in which the genetic parameters of the GEM are not directly observed (see Supplementary Notes). GEMs have previously modeled genetic variation at the resolution of gene presence-absence[20–23], but have not yet been used to link nucleotide-level genetic variation (i.e., alleles) to observed phenotypes (i.e., AMR) in a predictive manner[24]. Since alleles are the primary forms of causal variation identified in GWAS, an approach for mechanistically integrating information about alleles is of major interest[25].

Here we develop a GEM-based machine learning framework for modeling datasets used in GWAS and apply it to a sequencing dataset of drug-tested TB strains. We show that our framework achieves high performance in accurately classifying AMR phenotypes of TB strains. We then characterize the identified classifiers for pyrazinamide, isoniazid, and para-aminosalicylic acid AMR and show that they identify key genetic determinants and pathway activity discriminating between resistant and susceptible TB strains. This work demonstrates how GEMs can be used directly as an input-output machine learning model to extract both genetic and biochemical network-level insights from microbial GWAS datasets.

## Results

### Assessing AMR mechanisms motivates metabolic model approach

. We first set out to assess the scope of a potential mechanism-based genotype-phenotype map using a dataset of 1595 drug-tested TB strains[2,26] and a GEM of TB H37Rv, named iEK1011[27]. The acquired genetic variant matrix (**G**) of the 1595 strains describes 3739 protein-coding genes and their 12,762 allelic variants, where each variant is defined as a unique amino acid sequence for the protein coding gene. Our analysis therefore does not account for synonymous amino acid changes and intergenic genetic variants. The corresponding drug susceptibility status for a strain is described by a binary 'susceptibility' or 'resistance' phenotype to a particular antibiotic. iEK1011 accounts for 1011 genes (26% of H37Rv) and comprises a metabolic network of 1229 reactions and 998 metabolites.

Comparing the gene list between iEK1011 and the genomics dataset, we found that 26% (981/3739) of the total genes and 25% (3310/12,762) of the total variants described by the genetic variant matrix were accounted for by the GEM. To evaluate iEK1011's potential to model causal variants, we compiled a list of AMR genes and compared this list to the gene list of iEK1011 (Supplementary Data 1; Methods). We found that 72% (32/44) of known AMR genes are accounted for in iEK1011 (Supplementary Table 1). In the case of six drugs (ethambutol, isoniazid, d-cycloserine, para-aminosalicylic acid, ethionamide, and pyrazinamide), 87% (20/23) of their AMR genes were accounted for in iEK1011. AMR genes not explicitly accounted for in iEK1011 were primarily related to DNA transcription (e.g., *rpoB*) and transcriptional regulation (e.g., *embR*). The antibiotics rifampicin, ofloxacin, and streptomycin do not have AMR genes accounted for in iEK1011 and are therefore out of scope for our study. Taken together, the abundance of AMR genes accounted for in iEK1011 motivated a GEM-driven analysis of the TB AMR dataset.

### A flux balance framework for classifying microbial genomes

. While we have shown that a GEM accounts for the majority of known genetic determinants of AMR in TB, computational methods do not exist for integrating a fine-grained description of allelic variation with GEMs to directly predict binary phenotypes (i.e., AMR susceptible/resistant classification). We thus set out to develop a GEM-based machine learning framework for analyzing the TB dataset. The developed method, named Metabolic Allele Classifier (MAC), takes the genome sequence of a particular TB strain as its input and classifies strains as either resistant or susceptible to a specific antibiotic (Fig. 1a). Specifically, the MAC is an allele-parameterized form of flux balance analysis[28,29] that represents a strain as a set of allele-specific flux capacity constraints and classifies AMR according to the optimum value attained by optimizing an antibiotic-specific objective.

We formulate the MAC within the flux balance analysis framework as follows,

$$H_{y,k} = \text{sign}\left(\max_{\boldsymbol{v}} \boldsymbol{c}_y^T \boldsymbol{v}_k + b\right) \quad \text{(Antibiotic-specific objective)}$$

$$s.\,t.$$

$$\boldsymbol{S}\boldsymbol{v}_k = 0 \quad \text{(Flux balance constraint)}$$

$$\boldsymbol{v}^{lb} \leq \boldsymbol{v} \leq \boldsymbol{v}^{ub} \quad \text{(Over-all min/max flux constraints)}$$

$$\boldsymbol{G}\boldsymbol{a}^{lb} = \boldsymbol{v}^{lb} \leq \boldsymbol{v} \leq \boldsymbol{v}^{ub} = \boldsymbol{G}\boldsymbol{a}^{ub} \quad \text{(Allele-specific min/max flux constraints)}$$

$$(1)$$

Where each line of the MAC formulation in Eq. (1) is briefly described with plain text to the right, and further detailed by the correspondingly ordered bullet points below;

- $H_{y,k}$ is the sign of the MAC optimum value that classifies a strain, $k$, as either resistant (R) or susceptible (S) to a specific antibiotic, $y$ (see Supplementary Notes for comparison between the MAC and the Support Vector Machine). The optimum value is determined by optimizing the objective

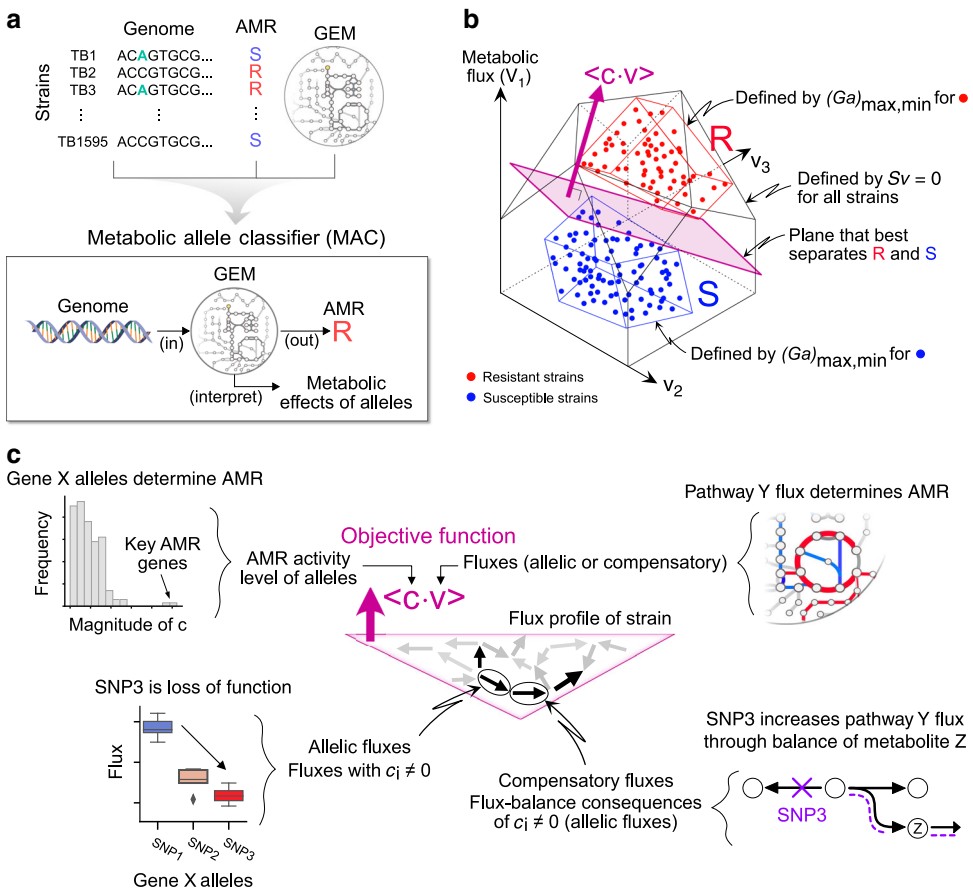

**Fig. 1 A metabolic systems approach for genetic associations. a** In this study, data describing TB genome sequences and AMR data types are integrated with a metabolic model to learn a biochemically-interpretable classifier, named Metabolic Allele Classifier (MAC). The MAC parameters consist of allele-specific flux capacity constraints, **a**, and an antibiotic-specific metabolic objective, **c**, both of which are inferred from the data. **b** The optimal MAC describes strain-specific polytopes in flux space that separate into resistant (R) and susceptible (S) regions. The MAC objective function, $c^Tv$, is identified as normal to the plane that best separates R and S. **c** The learned MAC provides a biochemically-based hypothesis of AMR mechanisms and allele-specific effects through interpretation of **c** and **v**. The genome-scale flux state of a strain, **v**, consists of fluxes that are directly activated by alleles (allelic fluxes) and those that are flux-balance consequences of the allele-activated fluxes (compensatory fluxes). Abbreviations: S, susceptible; R, resistant; AMR, antimicrobial resistance.

function, $\max c^T_y v_k$, which describes a linear combination of the metabolic fluxes, $v_k$, and is specific to an antibiotic, $y$. The antibiotic-specific objective coefficients, $c_y^T$, are unknown a-priori and inferred from the data as a normal to the plane that best separates resistant and susceptible strains (Fig. 1b).

- The classical flux-balance constraints, $Sv_k = 0$, ensure that for each strain, $k$, the net mass flux through each of their metabolites is balanced to 0 (i.e., steady internal homeostatic state), where $S$ is the stoichiometric matrix with 998 metabolites (rows) and 1229 reactions (columns).
- The constraints on the fluxes (reaction rates) through the metabolic reactions, $v^{lb,ub}$, describe the overall min/max flux constraints not changed by allelic variation and are thus the same for all strains. Geometrically, the constraints $v^{lb,ub}$ and $Sv_k = 0$ define a polytope in which all strain-specific fluxes must reside (Fig. 1b).
- The binary genetic variant matrix, $G_{k,i}$, is the primary data type used in GWAS and describes the presence/absence of $i$ alleles (columns) across $k$ strains (rows).
- The constraints, $G_{k,i}a_{i,j}^{lb,ub} = v_{k,j}^{lb,ub}$, represent the genome sequence of each strain (represented as a row in $G$) as a set of allele-specific flux constraints, $v_{k,j}^{lb,ub}$. The allele-constraint matrix, $a^{lb,ub}$, describes the allele-specific flux constraint values of $i$ alleles (rows) that encode for enzymes catalyzing $j$

reactions (columns) (see Supplementary Notes for further explanation on the biological relationship between alleles and flux constraints). The allele-constraint matrix is unknown a-priori and inferred from the data. Geometrically, $Ga$ describes strain-specific polytopes that represent the best separation of resistant and susceptible strains within the overall flux space (Fig. 1b).

Importantly, the MAC was formulated such that for each strain-antibiotic classification, $H_{y,k}$, there exists a corresponding flux state, $v_k$, thereby providing a biochemical network explanation of the classification. Geometrically, the flux state of the metabolic network of a particular strain is described by the intersection of the objective function with its genome-specific polytope (Fig. 1c).

The objective function corresponds to the fluxes through a set of metabolic reactions that form the basis for the MAC. By the fundamental nature of flux balancing, these reactions identify activity levels of discriminating pathways. The objective function that best separates the two polytopes formed by the spaces of resistant and sensitive phenotypes is a plane that describes a critical level of pathway activity that discriminates between the R and S phenotypes. Thus, the separating plane consists of fluxes that are directly activated by alleles ($c_i \neq 0$) and those that result

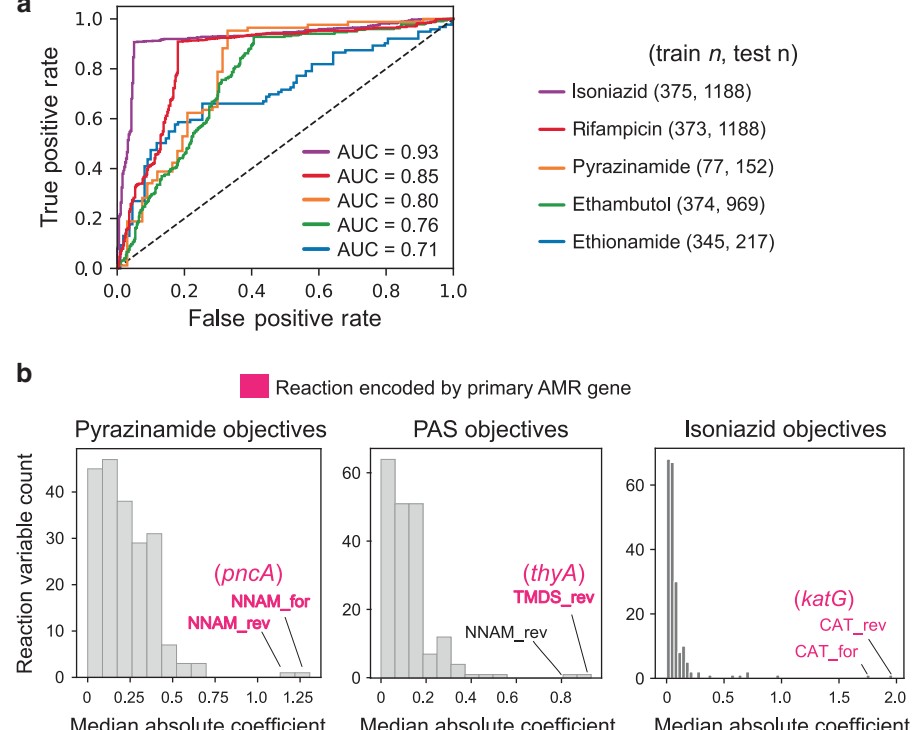

**Fig. 2 Validation of metabolic allele classifiers. a** Receiver operator characteristic (ROC) curves for MAC AMR predictions determined using a test set of 1188 isoniazid-tested strains. **b** Histogram of median absolute MAC objective function coefficients ($c_y^T$) for pyrazinamide, para-aminosalicylic acid, and isoniazid MACs. The reaction variables corresponding to the two largest coefficients are noted in text. The reaction variable corresponding to the primary genetic determinant is colored pink. Abbreviations: AUC, area under the curve.

from flux-balance consequences of $c_i \neq 0$. Statistical tests can then be performed using the set of all strain-specific intersections to identify both significant flux states discriminating between resistant and susceptible strains (Supplementary Fig. 2a) as well as their underlying allele-specific flux effects (Supplementary Fig. 2b). The MAC is therefore a biochemically interpretable machine learning classifier.

**Validation of metabolic allele classifiers**. We utilized randomized sampling, machine learning, and model selection to identify predictive MACs (see Supplementary Fig. 4-5, Methods, and Supplementary Notes for further details of the process outlined below). Specifically, the MACs were trained on the same 375 strains to predict antibiotic phenotypes with 1220 strains set aside for testing. Since the computational cost of estimating MACs scales poorly with the number of alleles utilized, we limited the set of alleles modeled by the MAC to 237, describing 107 genes consisting of both known and unknown relations to AMR (Supplementary Data 1). The known AMR genes provide validation cases while the unknown genes enable novel insights.

We assessed MACs for isoniazid, rifampicin, pyrazinamide, ethambutol, and ethionamide using held out test sets and find that the MACs generally achieve high classification performance (Fig. 2a), with scores similar to our previous mechanism-agnostic machine learning models[2]. The MACs were further validated by assessing their ability to recover the primary AMR genes. We find that the largest objective weights for pyrazinamide, para-aminosalicylic acid, and isoniazid MACs correspond to the primary known AMR genes of antibiotics (Fig. 2b). These results show that the MAC performs on par with state-of-the-art machine learning approaches in AMR classification and identification of primary AMR genes.

**MACs reveal known and new antibiotic resistance determinants**. The ability of MACs to efficiently predict AMR phenotypes (i.e., high accuracy, low complexity) suggests that the model parameters have biological relevance. Furthermore, in contrast to black-box machine learning models, the genotype-phenotype map of a MAC was designed to satisfy known biological constraints on metabolism e.g., reaction stoichiometry, mass conservation, gene-product-reaction encoding, nutrient environment. Therefore, we hypothesized that MACs should not only identify genetic determinants of AMR, but also provide metabolic systems explanations of their predictions.

Below, we focus our analysis on three case studies: pyrazinamide, para-aminosalicylic acid, and isoniazid AMR. These three antibiotics were chosen due to having both characterized and uncharacterized mechanisms underlying their associated alleles, allowing for both test cases and novel insights for the MAC. We analyze the best MACs for each antibiotic through four steps: (i) identification of significant fluxes discriminating between resistant and susceptible strains (i.e., flux GWAS), (ii) pathway enrichments of significant fluxes, (iii) identification of key allelic flux effects, and (iv) network-level flux tracing of allelic effects (Methods).

**MACs for pyrazinamide resistance**. To identify key flux states discriminating between resistant and susceptible strains, we performed statistical associations between the strain-specific MAC fluxes, $v_k$, and pyrazinamide AMR phenotypes using the training set of 77 strains (52 resistant, 25 susceptible) (we refer to this as Flux GWAS, see Fig. 1d). Flux GWAS identified 25 significant reaction fluxes (Bonferroni corrected $P < 4.66 \times 10^{-5}$, 0.05/1073 reactions) whose gene–protein reaction rules overlapped with 8 genes modeled by MAC alleles (*pncA, ansP2, fadD26, ppsA,* and *drrABC*) (Supplementary Fig. 7a; Supplementary Data 3).

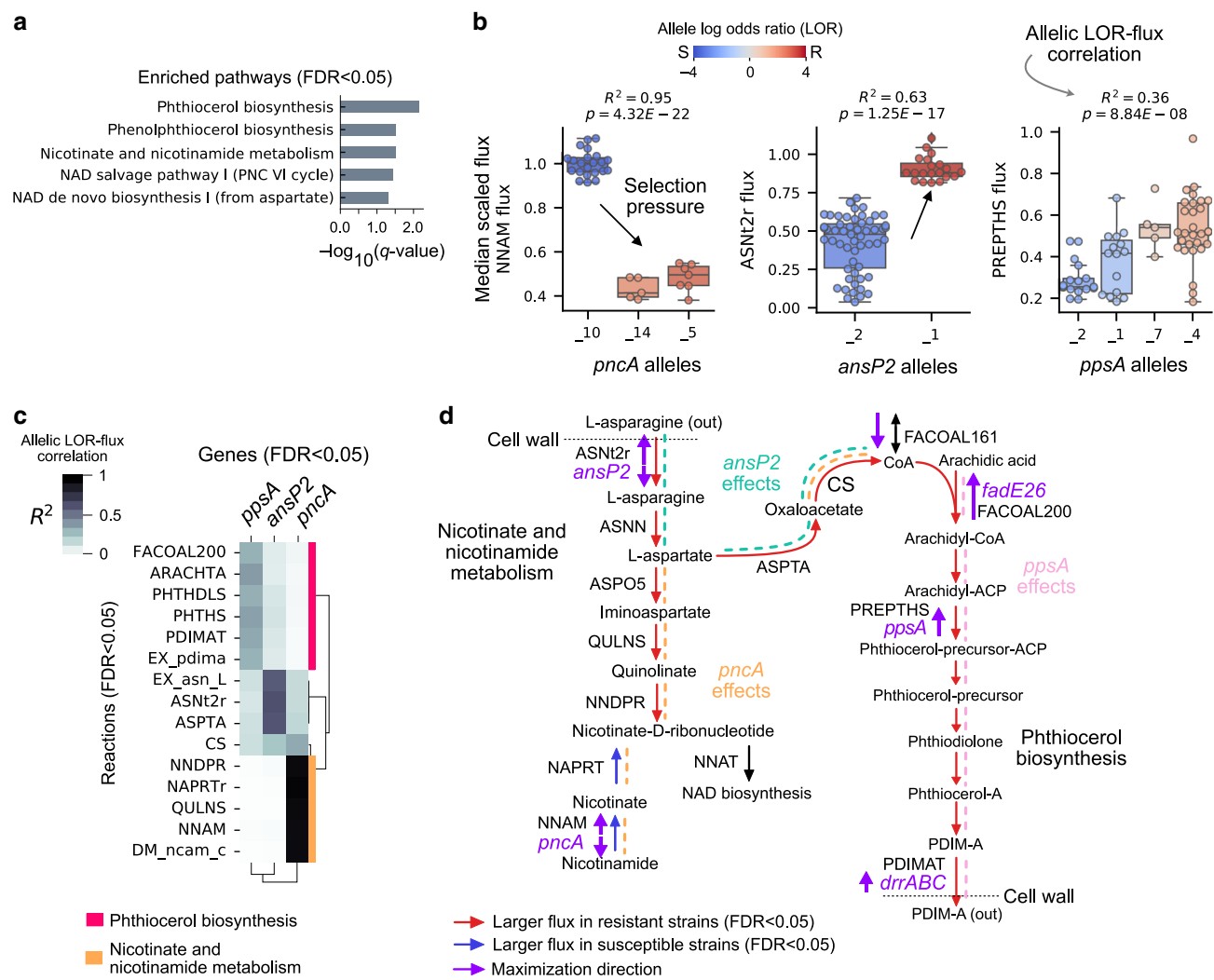

**Fig. 3 Characterization of pyrazinamide MACs. a** Horizontal bar plots of pathways enriched with significant pyrazinamide-associated fluxes with FDR < 0.05. **b** Boxplots of *pncA*, *ansP2*, and *ppsA* allele-specific fluxes for the reactions catalyzed by their gene-products. Alleles are rank ordered from least to greatest by their log odds ratio (LOR), from left to right. The boxes are colored according to the allele LOR, where positive corresponds to resistant (R) dominant while negative corresponds to susceptible (S) dominant. The box shows the quartiles of the dataset with the median noted as the horizontal line while the whiskers extend to show the rest of the points with outliers colored gray. The *R*-squared and *P*-value for the regression between allele LOR and flux is noted. See Supplementary Data 3 for list of mutations per allele. **c** Clustered heatmap of allele LOR-flux correlations between genes (*y*-axis) and significant reactions fluxes (*x*-axis). **d** Pathway depiction of nicotinate and nicotinamide metabolism and phthiocerol biosynthesis with objective variables plotted. Coenzyme-A generation from L-asparagine through aspartate decarboxylase (ASPTA) and citrate synthase (CS) is also depicted. Traced allelic effects are shown as dashed lines and colored for *pncA*, *ansP2*, and *ppsA*.

To gain a coarse systems view of the 25 significant fluxes, we performed pathway enrichment tests using a curated gene-pathway annotation list consisting of both BioCyc[30] and KEGG pathways[31] that accounts for 32% of protein-coding genes in the H37Rv genome (1254/3906) (Supplementary Data 2; Methods). Of the 245 total pathways, 5 were enriched with significant fluxes with less than 5% false discovery rate (FDR < 0.05)[32] and were primarily described by phthiocerol biosynthesis and nicotinate and nicotinamide metabolism (Fig. 3a). These results recapitulate two pyrazinamide features describing flux variation in nicotina-midase activity[33] and phthiocerol dimycocerosate (PDIM) biosynthesis[12].

We then set out to understand the genetic basis for the flux associations by identifying loci in which the AMR association of each allele was correlated with their flux distribution (LOR-flux correlation) (see Methods). The idea here is that resistant alleles have different metabolic effects than susceptible alleles for key genes. These allele-specific flux differences underlie the AMR

classification accuracy of the MAC. We identified significant LOR-flux correlations at *pncA*, *ansP2*, and *ppsA* loci (FDR < 0.05) (Fig. 3b). Specifically, the MACs infer a flux decreasing selection pressure at the *pncA* locus and flux increasing selection pressures at the *ansP2* and *ppsA* loci. The estimated decreased enzymatic activity of *pncA* is consistent with studies describing resistant *pncA* mutants as loss of function[34]. Mutations in *ppsA* have previously been linked to pyrazinamide AMR[12] and convergent AMR evolution[35] while *ansP2* mutants have not yet been associated with AMR.

To understand the global effects of *pncA*, *ppsA*, and *ansP2* alleles on the metabolic network, we traced out their LOR-flux correlation through the 25 significant reactions (Fig. 3c). For *ansP2*, we observe that the increased generation of L-asparagine by the resistant *ansP2* allele was utilized to generate coenzyme A (CoA) through aspartate aminotransferase (ASPTA) and citrate synthase (CS) (Fig. 3d), which recapitulates experimental studies describing L-aspartate-based modulation of CoA as a pyrazinamide resistance

mechanism[12]. However, our results differ from that of the proposed *panD*-based pantothenate route for CoA generation[36–38]. The lack of pyrazinamide-associated *panD* alleles in our dataset may underlie this discrepancy.

In summary, pyrazinamide MACs correctly identify *pncA* and *ppsA* alleles as major genetic determinants and recapitulate nicotinamide metabolism, CoA biosynthesis, and phthiocerol metabolism as key metabolic associations[12,34]. As for new hypothesis, the MACs implicate *ansP2* mutants in resistance through L-aspartate-based modulation of the coenzyme-A pool.

**MACs for Para-aminosalicylic acid resistance**. We performed flux GWAS using the para-aminosalicylic acid training set of 375 strains (80 resistant, 295 susceptible) and identified 52 fluxes discriminating between resistant and susceptible strains (Bonferroni corrected $P < 4.66 \times 10^{-5}$, 0.05/1073 reactions) (Supplementary Fig. 7b, Supplementary Data 4). Of these 52 reactions, 10 were directly encoded by MAC alleles of 8 genes (*thyA*, *katG*, *pncA*, *alar*, *cysK2*, *ald*, *fadE26*, *aspB*, *kdg*, and *inhA*). Pathway enrichment tests of these 52 reactions identified S-adenosyl-L-methionine cycle II, NAD de novo biosynthesis I (from aspartate), and cysteine and methionine metabolism as key para-aminosalicylic acid pathways (FDR < 0.05) (Fig. 4a). The identification of cysteine and methionine metabolism recapitulates known metabolic effects of para-aminosalicylic acid[39].

We tested these genes for allelic LOR-flux correlations and identified selection pressures at *thyA*, *cysK2*, *alr*, *pncA*, and *fadD26* loci (FDR < 0.05, $R^2 > 0.1$) (Fig. 4b). Specifically, the MACs infer flux decreasing selection pressures at the *thyA*, *cysK2*, *pncA*, and *fadD26* loci and a flux increasing selection pressure at the *alr* locus. The estimated decreased enzymatic activity of *thyA* resistant alleles is consistent with experimental studies describing *thyA* resistant mutants as loss of function[8,40]. The identification of *alr* and *pncA*—known determinants of cycloserine and pyrazinamide, respectively—reflect the co-resistance of these strains and are not known to have selective pressure in para-aminosalicylic acid treatment. Of these genes, only *cysK2* encodes an enzyme in cysteine and methionine pathway and has not been previously linked to AMR.

We traced out the allelic LOR-flux correlation of *cysK2* through cysteine and methionine pathway flux and found that their effects positively correlated with *fadD26* alleles and negatively with *thyA*, *alr*, and *pncA* alleles (Fig. 4c). Resistant *cysK2* alleles are estimated to lead to increased flux through O-succinylhomoserine (SHSL2r) and cystathionine beta-synthase (CYSTS). The effect of *cysK2* decreases from SHSL2r to CYSTS at the L-homocysteine flux balance node, which implicates L-homocysteine modulation as the *cysK2* selection pressure (Fig. 4d). Notably, L-homocysteine was experimentally identified as the most differentially perturbed metabolite resulting from para-aminosalicylic acid treatment[39].

In summary, para-aminosalicylic acid MACs recover *thyA* as the primary genetic determinant and recapitulate cysteine and methionine metabolism as a major pathway induced by the drug. As for novel hypothesis, the MACs implicate deleterious *cysK2* mutants in resistance through modulation of L-homocysteine that may either arise from deleterious *thyA* mutants or para-aminosalicylic acid treatment.

**MACs for isoniazid resistance**. We performed flux GWAS using the isoniazid training set of 375 strains (248 resistant, 127 susceptible) and identified 160 significant fluxes (Bonferroni corrected $P < 4.66 \times 10^{-5}$, 0.05/1073 reactions) (Supplementary Fig. 7c, Supplementary Data 5). We find that only 11.3% (18/160) of the significant fluxes were catalyzed by gene-products of the MAC alleles. Pathway enrichments of the 160 significant fluxes

identified TCA cycle V, oxidative phosphorylation, superpathway of mycolate biosynthesis, and gluconeogenesis I as key isoniazid pathways (FDR < 0.05) (Fig. 5a). These results are consistent with numerous studies demonstrating TCA and oxidative phosphorylation as key TB pathways altered by isoniazid treatment[41–43] and studies generally linking antibiotic efficacy to these pathways[44]. In general, we found that resistant strains were characterized by decreased respiratory activity, which is consistent with studies connecting decreased respiration to increased isoniazid resistance[42]. The genes encoding enzymes in these enriched pathways correspond to known (*inhA*, *fabD*, *kasA*, *accD6*, *fadE24*, *ndh*) and unknown (*accD5*, *nuoL*, *gpdA2*) genetic determinants of isoniazid resistance; however, none of these encoded for reactions annotated with TCA cycle V.

We tested the significant fluxes for allelic LOR-flux correlations and identified selection pressures at *katG*, *ndh*, *nuoL*, *accD6*, *gpdA2*, *fabD*, *kasA*, and *accD5* loci (FDR < 0.05) (Fig. 5b). Specifically, the MACs infer flux decreasing selection pressures at the *ndh*, *nuoL*, *fabD*, *gpdA2*, and *kasA* loci and a flux increasing selection pressure at the *katG*, *accD6*, and *accD5* locus (MCOATA is depicted in reverse direction). The resulting increased CAT flux observed in resistant strains is consistent with studies describing the majority of resistance-conferring *katG* alleles in clinical isolates as preserving catalase-peroxidase activity while disabling isoniazid binding (i.e., strains carrying susceptible-dominant *katG* alleles have low catalase-peroxidase flux due to isoniazid binding)[45,46]. The increased flux towards mycolic acid biosynthesis in resistant strains by *fabD*, *accD6*, and *kasA* is consistent with studies showing increased expression of these genes resulting from isoniazid treatment[47]. Furthermore, the metabolite acted on by these genes, malonyl-CoA, has recently been shown to have a significant fold change in response to 16 antibiotics in TB[48].

We traced out significant LOR-flux correlations of these genes through the enriched pathways to elucidate their global network effects (Fig. 5c). For the novel genetic determinants, *nuoL* and *gpdA2*, we find that their alleles have significant flux effects in cytochrome bd oxidase reactions (CYTBD, CYTBD2) traced through menaquinone and ubiquinone flux balance nodes, respectively (Fig. 5d). The allelic effects of the primary genetic determinant, *katG*, are similarly traced through cytochrome bd oxidase flux by oxygen. The importance of cytochrome bd oxidase has recently been linked to isoniazid[41]. These results implicate *gpdA2* and *nuoL* mutants in isoniazid AMR through modulation of quinone/menaquinone pools.

In summary, isoniazid MACs recover the primary (*katG*) and secondary (*inhA*, *fabD*, *kasA*, *accD6*, *fadE24*, *ndh*) genetic determinants and recapitulate oxidative phosphorylation, TCA, and mycolic acid biosynthesis as major pathways induced by the drug[41–43]. As for novel genetic hypothesis, the MACs implicate *gpdA2* and *nuoL* mutants in resistance through modulation of menaquinone and ubiquinone that may either arise from *katG* mutants or isoniazid-induced oxidative stress.

**Conventional pathway analyses do not recapitulate mechanisms**. To assess how MAC results compare to mechanism-agnostic approaches, we performed conventional pathway analysis of the 197 alleles (Supplementary Data 6, Methods). Comparison of pathway-based analysis showed that results derived from conventional pathway enrichments do not recapitulate the antibiotic mechanisms for isoniazid, pyrazinamide, and para-aminosalicylic acid. For isoniazid, a total of five pathways were enriched (FDR < 0.05); however, the significant allelic associations enriched in pathways were simply those annotated for *katG*, such as superoxide radicals degradation and tryptophan metabolism.

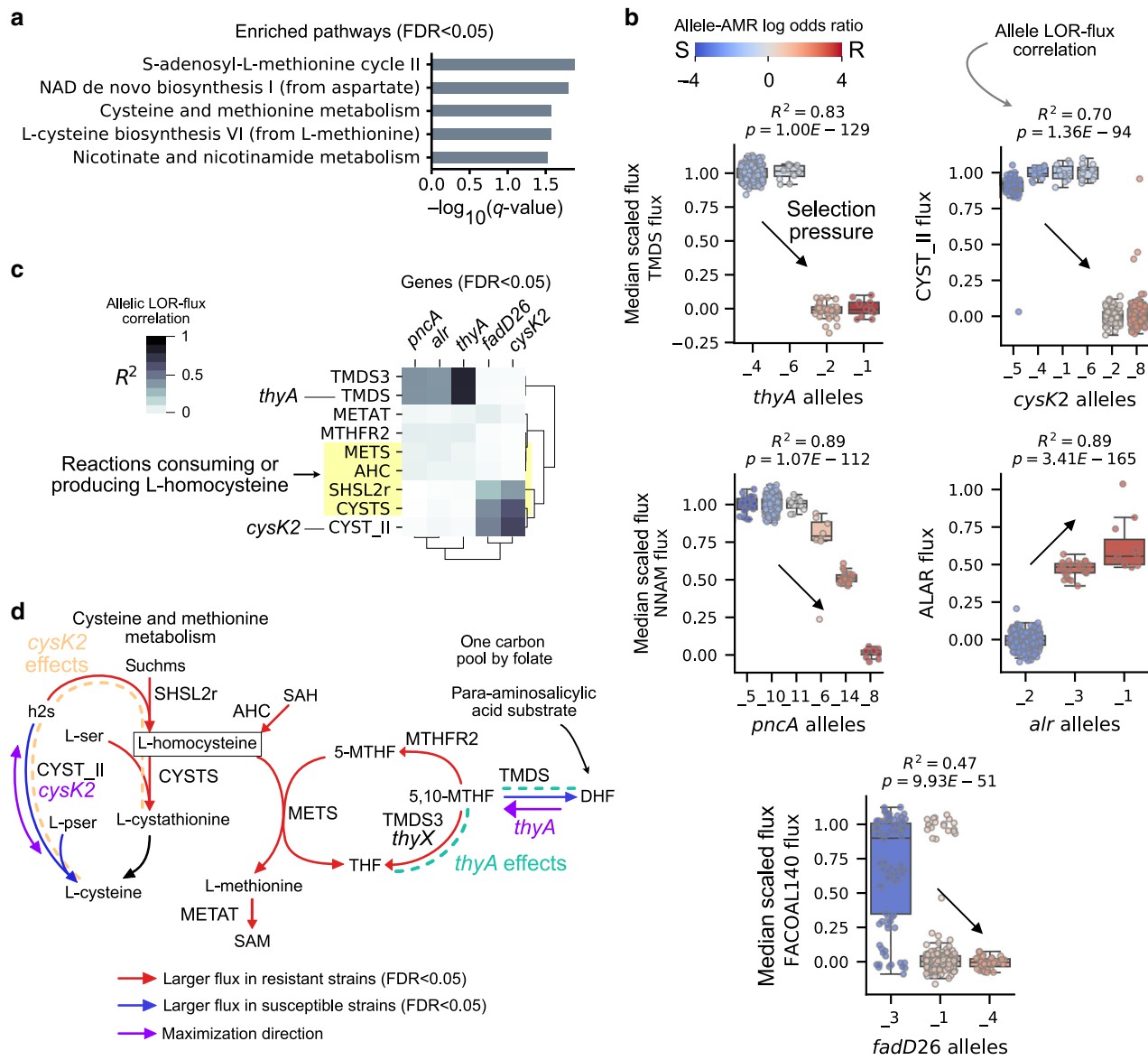

**Fig. 4 Characterization of para-aminosalicylic acid MACs. a** Horizontal bar plots of pathways enriched with significant para-aminosalicylic acid-associated fluxes with FDR < 0.05. **b** Boxplots of *thyA*, *cysK2*, *alr*, *pncA*, and *fadD26* allele-specific fluxes for the reactions catalyzed by their gene-products. Alleles are rank ordered from least to greatest by their log odds ratio (LOR), from left to right. The boxes are colored according to the allele LOR, where positive corresponds to resistant (R) dominant while negative corresponds to susceptible (S) dominant. The box shows the quartiles of the dataset with the median noted as the horizontal line while the whiskers extend to show the rest of the points with outliers colored gray. The *R*-squared and *P*-value for the regression between allele LOR and flux is noted. See Supplementary Data 4 for list of mutations per allele. **c** Clustered heatmap of allele LOR-flux correlations for significant reactions in cysteine and methionine metabolism. **d** Pathway depiction of cysteine and methionine metabolism and one carbon pool by folate. Significant allelic effects are shown by dashed lines and colored for *thyA* and *cysK2*.

For para-aminosalicylic acid, L-alanine biosynthesis I was the only enriched pathway while no pathway was enriched for pyrazinamide alleles (FDR < 0.05).

These results show that flux balance constraints are required to generate meaningful network-level hypotheses for identified genetic associations. The basis for this advancement is that flux balances represent how the entirety of metabolic gene products come together to produce balanced homeostatic states.

## Discussion

We have developed a computational framework for analyzing data sets (comprised of genotypes and binary phenotypes) using a genome-scale model (GEM) to identify the genetic and metabolic basis for TB AMR (Fig. 1a). The identification of the underlying biochemical mechanisms is reflected in the MAC. We first discuss our approach, emphasizing key design choices, and then describe the results it generates when applied to the TB dataset.

The outcome of the MAC depends on two major design choices: the set of alleles and the objective function that optimally separates strains into resistant and sensitive strain cohorts in the overall metabolic flux space. Although our approach does not explicitly require prior knowledge of key AMR genes, we chose a set of alleles with just over 100 genes with known and implicated AMR relations in order to both provide test cases and to address the combinatorial explosion of sampling possible allelic effects. Relaxing the current computational bottleneck in identifying MACs will enable the utilization of all alleles. For determining the

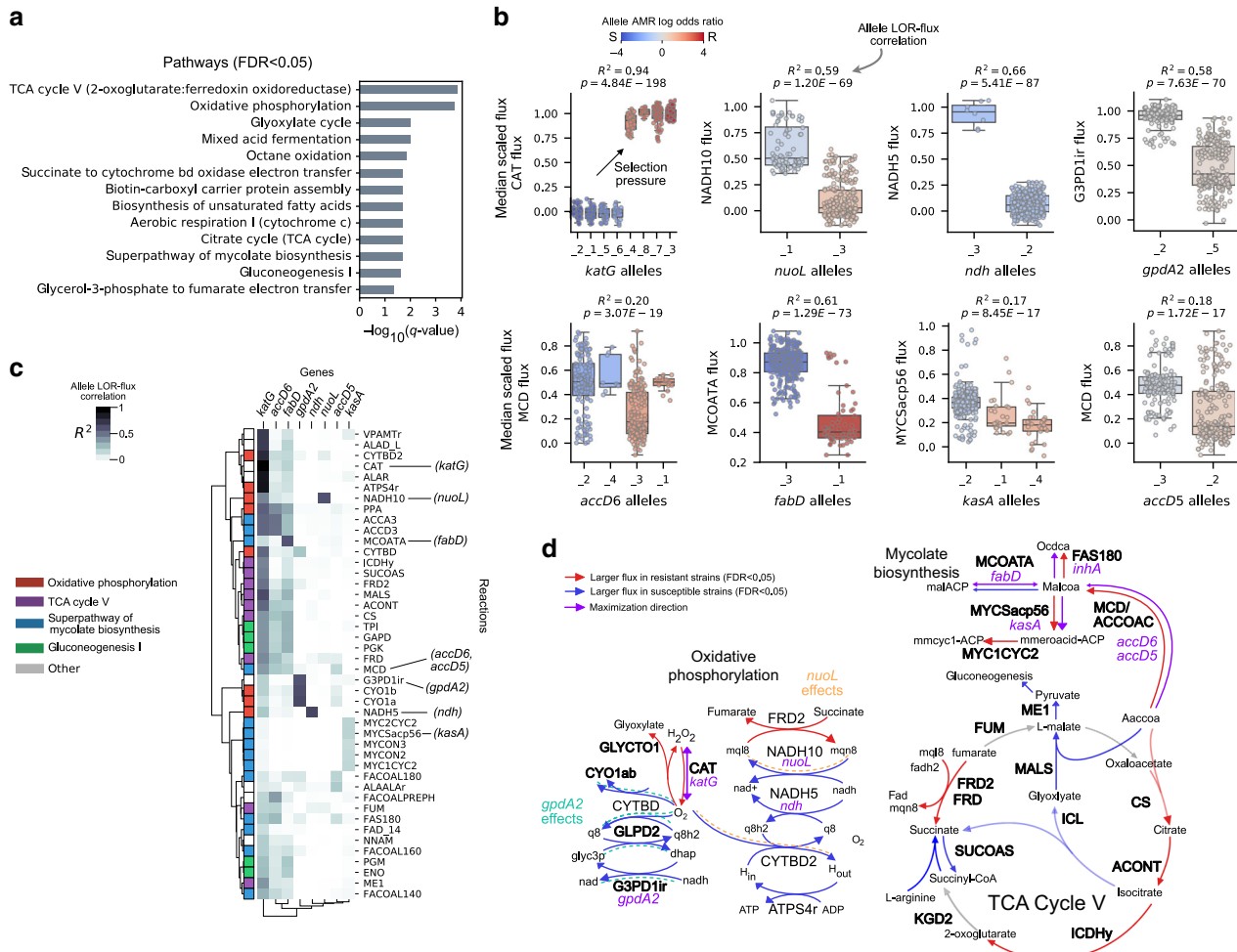

**Fig. 5 Characterization of isoniazid MACs. a** Horizontal bar plots of pathways enriched with significant isoniazid-associated fluxes with FDR < 0.05.
**b** Boxplots of *katG*, *nuoL*, *ndh*, *gpdA2*, *accD6*, *fabD*, *kasA*, and *accD5* allele-specific fluxes for the reactions catalyzed by their gene-products. Alleles are rank ordered from least to greatest by their log odds ratio (LOR), from left to right. The flux (y-axis) is the median scaled flux across the high-quality isoniazid MACs. The boxes are colored according to the allele LOR, where positive corresponds to resistant (R) dominant while negative corresponds to susceptible (S) dominant. The box shows the quartiles of the dataset with the median noted as the horizontal line while the whiskers extend to show the rest of the points with outliers colored gray. The *R*-squared and *P*-value for the regression between allele LOR and flux is noted. See Supplementary Data 5 for list of mutations per allele. **c** Clustered heatmap of allele LOR-flux correlations for significant reactions in TCA Cycle V, Oxidative phosphorylation, and Mycolate biosynthesis. **d** Pathway depiction of TCA Cycle V, Oxidative phosphorylation, and Mycolate biosynthesis. Significant allelic effects are shown by dashed lines and colored for *gpdA2* and *nuoL*.

objective function, our approach was based on the key insight that a linear program may behave as a machine learning classifier if its objective optimizes in the direction normal to a predictive classification plane. While we utilized PCA, L1-logistic regression, and the BIC metric to identify sparse linear objectives, there are potentially alternative avenues that could be taken. The major concept that should sustain in any model selection strategy is that a good model is simple (in structure) yet accurate (in its predictions). Application of the MAC to other GWAS datasets may therefore benefit from tuning these parameters appropriately.

The MAC advances current GWAS machine learning approaches by enabling a biochemical interpretation of genetic associations. Although advancements have been made to increase the explainability of black-box machine learning models[49–51], such interpretations are limited by the lack of mechanistic knowledge incorporated in the model. We show that causal biochemical explanations for classifications can be derived by constraining a machine learning classifier to satisfy knowledge-based biological constraints (gene function, reaction stoichiometry, flux balance, etc).

Our interpretation of MACs for pyrazinamide, para-aminosalicylic acid, and isoniazid AMR identified genome-scale flux states and key pathways discriminating resistant and susceptible strains. Notably, we found the MAC-identified pathways to be consistent with known antibiotic mechanisms. In contrast, conventional pathway analysis using only alleles was unable to recapitulate known pathway mechanisms. The MAC therefore provides a mechanistic approach for pathway-based analysis of genome-wide associations[52].

Dissection of the allele-specific fluxes underlying the significant fluxes further clarified the genotype-phenotype map and provided hypotheses regarding specific allelic effects. For example, pyrazinamide MACs implicate an *ansP2* allele as a novel resistance determinant through increased uptake of asparagine towards L-aspartate-based CoA generation. The MAC thus extends allele-phenotype associations (i.e., LOR) by estimating allele-specific flux effects and their network interactions.

Taken together, the framework presented here meets the pressing need to integrate comprehensive biochemical mechanisms for the analysis of genomics-phenomics datasets. Our

framework both recovers known gene-AMR relations and provides novel insights regarding their metabolic basis. As genome sequences, phenotypes, and genome-scale network reconstructions of microbes continue to grow in size and scope, similar results to those presented here are likely to appear in the coming years. This initial development of an FBA based GWAS analysis (FBA-GWAS) is likely to continue the development of a mechanistic basis into future GWAS methods.

## Methods

**Characteristics of utilized datasets.** The TB AMR datasets utilized in this study were acquired from a previous study that performed machine learning and protein structure analysis. References describing this data set are provided in the supplementary information of the previous study[2]. The dataset was initially acquired from the PATRIC database[26]. The sequencing and phenotypic testing data for these strains were generated at the Broad Institute. Additional information for these sequencing projects can be found at the Broad Institute website for the TB Antibiotic Resistance Catalog (TB-ARC).

**Curation and functional assessment of TB AMR genes.** A list of known and implicated TB AMR genes was curated for 8 antibiotics (isoniazid, rifampicin, ethambutol, pyrazinamide, ofloxacin, d-cycloserine, para-aminosalicylic acid) using a combination of databases[53], experimental studies, and computational studies[2,35,54,55]. Experimental studies on allele-specific effects for these AMR genes were curated utilizing a previous study performing 3D structural mutation mapping[2] and functional annotation from UNIPROT[56]. The lists of known and implicated TB AMR genes and mutational effects are provided (Supplementary Data 1).

**Modification of base genome-scale model.** We performed minor modifications to the base genome-scale model, iEK1011, in order to use it for the MAC. Specifically, we performed quality-assurance and quality check (QA/QC) by removing blocked reactions (i.e., cannot carry any flux) and imposing maximum and minimum allowable flux constraints on the model determined by Loopless Flux Variability Analysis (LFVA)[57,58]. Before FVA-derived constraints were imposed, we parameterized the exchange reactions according to the experimental nutrient media for testing AMR phenotypes, Middlebrook 7H10 (m7H10). Specifically, the LFVA simulations were constrained to a biomass flux of at least 10% of its maximum value, and the total flux was bounded from above by 1.5 times the minimum total flux determined by parsimonious flux balance analysis[59]. The code for initializing the base genome-scale model is provided in the code repository.

**Discretization of flux solution space for allelic effects.** We determined the set of potential constraints imposed by an allele through discretization of the flux solution space. Following QA/QC, the flux solution space was sampled by Markov-Chain Monte-carlo sampling[60,61], resulting in a probability distribution for each reaction flux. The solution space can then be discretized by first splitting the flux space in half at the mean flux. The upper bounds are then constructed by taking equal intervals from the mean to the maximum upper flux. The lower bounds are constructed similarly by setting them at equal intervals from the mean to the minimum lower flux. Notably, this discretization of the sampled flux space into upper and lower bound constraints requires the explicit definition of the total number of potential constraints an allele can potentially be mapped to. Specifying the set of constraints per allele determines the possible flux variation in the population. The set of constraints per allele was chosen to be minimally sufficient in our case, owing to the coarse resolution of the binary AMR phenotypes. Increasing the number of constraints per allele provides a finer resolution of the flux solution space, but comes at the cost of increasing the number of sampled MACs. We tested variations of the discretization resolution and found that while more constraints generally allow for the largest variety of variation, increasing the number of potential constraints by a linear factor leads to an exponential increase in the size of the solution space, requiring more samples. We found 4 to 10 constraints to be sufficient for generating popFVA states capable of explaining observed phenotypic variation. All constraint-based modeling was performed using the cobrapy package version 0.15.3[62].

**Randomized sampling of allele-constraint map ensemble.** Since knowledge of allele-specific effects are unavailable, we generated an ensemble of landscapes through randomized sampling of the allele-constraint map. Specifically, we generated an allele-constraint sample by sampling from each allele's discretized constraint set. The constraint set per allele includes the no change option and has a uniform probability distribution (i.e., each constraint has equal probability). An allele-constraint map sample is thus derived from sampling each allele's constraint distribution for all alleles.

**Estimation of MAC objective function.** In order to identify the antibiotic-specific objective functions for each MAC, we first comprehensively evaluated the metabolic consequences of their allele-constraint map sample using a population extension of Flux Variability Analysis (FVA)[63], named population FVA (popFVA). The popFVA linear program is formulated as follows,

$$\max_{v}/\min_{v} v_{k,j} \quad \text{(Maximize and minimize flux through all } j \text{ reactions)}$$
$$s.t.$$
$$\boldsymbol{S v_k} = 0 \quad \text{(Flux balance constraint)} \quad (2)$$
$$\boldsymbol{v^{lb}} \le \boldsymbol{v} \le \boldsymbol{v^{ub}} \quad \text{(Over-all min/max flux constraints)}$$
$$\boldsymbol{Ga^{lb}} = \boldsymbol{v^{lb}} \le \boldsymbol{v} \le \boldsymbol{v^{ub}} = \boldsymbol{Ga^{ub}} \quad \text{(Allele-specific min/max flux constraints)}$$

Where $G$ is the genetic variant matrix with $k$ strains and $i$ alleles, and where $S$ is the stoichiometric matrix. The matrices $\boldsymbol{a^{ub}}$ and $\boldsymbol{a^{lb}}$ describe the mapping of alleles to upper bound (ub) and lower bound (lb) flux constraints, respectively (allele-constraint map). Optimizing the minimum and maximum flux through all allele-catalyzed reactions represents our ignorance of the true evolutionary forces underlying the dataset. Once popFVA is computed for the MAC allele-constraint map, we then approximate the MAC linear objective using a series of steps described below.

1. We first use principal component analysis (PCA) to decompose the popFVA landscape, $X$, into a linear combination of two matrices, $U$ (strains, PCA components) and $V^T$ (popFVA features, PCA components) (e.g., X = UV). Prior to decomposition, $X$ was first normalized using minmax scaling. PCA was constrained to explain at least 90% of the total variation and implemented using the pca function in the scikit-learn toolbox v.0.20.3[64].

2. The $U$ matrix was then fit using L1-regularized logistic regression (LogReg) to predict AMR phenotypes. L1-regularized logistic regression was implemented using the Logit function in the statsmodels package version 0.9.0[65] with parameters maxiter, disp, and alpha set to 1500, False, and 0.5, respectively. An intercept was included for the regression model using the add_intercept function in statsmodels. The process of using PCA with regression is known as principal component regression (PCR).

3. To identify a linear programming objective from the PCR model, we make the key observation that the PCR function is a linear function normal to the decision boundary and has increasing/decreasing probability of classifying strain as resistant as you go further from the decision boundary (i.e., probability is closer to 0.5 at boundary and closer to 0 or 1 as you go further away). Therefore, the PCR function itself provides an ideal template for identifying a linear programming objective. We expect that the MAC predicts increasing or decreasing resistance as we maximize or minimize the objective value. Since the MAC objective function operates on the flux space, a series of mathematical transformations were taken to go from the PCR popFVA model to the MAC objective function (i.e., LogReg(PCA(FVA fluxes) → LogReg(v)). We start with the PCA decomposition of the popFVA fluxes (3)

$$\text{PCA(FVA fluxes)} \rightarrow \boldsymbol{U} = \boldsymbol{XZ} \quad (3)$$

Where $\mathbf{X}$ is the popFVA fluxes (strains, popFVA features), and $\boldsymbol{U}$ describes the PCA components (strains, PCA components) and $\boldsymbol{Z}$ has shape (popFVA features, PCA components). Fitting logistic regression to predict AMR using the PCA components gives the following Eq. (4)

$$\text{LogReg(PCA(FVA fluxes))} \rightarrow Y = b_0 + b_1 \boldsymbol{u}_1 + \cdots + b_k \boldsymbol{u}_k \quad (4)$$

Where $u$ describes the k PCA components and $b$ describes the LogReg coefficients. From this, we transform back to FVA space using the following mapping (5)

$$\boldsymbol{X} = \boldsymbol{UZ}^T \quad (5)$$

Which leads to the new LogReg equation,

$$\text{LogReg(FVA fluxes)} \rightarrow Y = b_0 + c_1 \boldsymbol{x}_1 + \cdots + c_m \boldsymbol{x}_m \quad (6)$$

Where $c_m = \sum_{i=1}^{k} z_{m,i} b_i$ for $m$ popFVA variables and $k$ PCA components. The $c$ values become the coefficients in the MAC objective function by representing the $V_{max}$ popFVA variables as $V_{forward}$ flux variables and $V_{min}$ popFVA features as $V_{reverse}$ flux variables.

If the objective is a minimization, then it is converted to a maximization by multiplying the objective function by −1.

**Assessment of MAC quality and model selection.** We used the Bayesian-Information criterion (BIC) to assess the quality of each MAC sample. Specifically, the BIC was derived from the PCR model used to infer the MAC objective. Since the BIC value, by itself, is not interpretable, high-quality MACs were determined according to their specific distance from the minimum BIC value, $\Delta BIC_i$ (i.e., $\Delta BIC_i = 0$ for minimum BIC model). We chose a $\Delta BIC_i$ cutoff of 10, which is in line with a rule of thumb that meaningful models (i.e., relatively high empirical support) should have $\Delta BIC_i < 10^{66}$.

**Flux GWAS**. We performed statistical tests to identify MAC fluxes significantly associated with AMR phenotypes. The goal was to determine which fluxes differentiate resistant and susceptible strains. Specifically, we tested whether the median scaled flux of a reaction was linearly correlated AMR phenotypes using an ANOVA F-test implemented by the f_classif function in the scikit-learn toolbox v.0.20.3[64]. The strain-specific reaction fluxes per MAC were normalized to be between zero and one using the MinMaxScaler function in the scikit-learn toolbox[64]. The set of significant reaction fluxes was determined by the Bonferroni-corrected significance threshold set at $P < 0.05/1073 = 4.66 \times 10^{-5}$.

**Pathway enrichments for significant fluxes**. We identified metabolic pathways enriched in significant AMR-associated fluxes through hypergeometric enrichment tests using the scipy function hypergeom[67]. The set of pathways was curated by combining gene-pathway annotations using both BioCyc[30] and KEGG[31] pathway annotations of TB genes. Pathways with 2 or less reactions were removed from the list, leaving a total of 264 pathways. We identified significant pathways as having less than 5% false discovery rate (FDR) correction by the Benjamini-Hochberg method.

**Statistical tests for allelic AMR and flux stratification**. We tested the AMR-based flux stratification of alleles by fitting a linear regression line between the allele log odds ratio (LOR) and fluxes. Linear regression was implemented using the linregress function in the scipy package. The LOR for each allele with respect to a specific antibiotic was quantified as $LOR = \log_{10}((PR/PS)/(AR/AS))$. PR, PS, AR, and AS denote number of strains that have the allele and are resistant (PR), have the allele and are susceptible (PS), do not have the allele and are resistant (AR), and do not have the allele and are susceptible (AS), respectively. If any of the values were 0, then 0.5 was added to each value to ensure a value when computing the logarithm. The fluxes for each allele were defined as the set of fluxes in strains containing that allele. We identified significant allelic LOR-flux correlations as having less than 5% FDR by the Benjamini-Hochberg method.

**Conventional pathway analysis of allelic variants**. We identified metabolic pathways enriched in the alleles of key AMR genes through hypergeometric enrichment tests using the scipy function hypergeom and the gene-pathway annotation list described above. We identified significant pathways as having less than 5% false discovery rate (FDR) correction by the Benjamini-Hochberg method.

**Reporting summary**. Further information on research design is available in the Nature Research Reporting Summary linked to this article.

## Data availability
The TB AMR datasets utilized in this study were acquired from a previous study that performed machine learning and protein structure analysis[2]. References describing this data set are provided in the supplementary information of the previous study[2]. The dataset was initially acquired from the PATRIC database[26]. The sequencing and phenotypic testing data for these strains were generated at the Broad Institute. Additional information for these sequencing projects can be found at the Broad Institute website for the TB Antibiotic Resistance Catalog (TB-ARC).

## Code availability
Code for MACs is available on GitHub (https://github.com/erolkavvas/metabolic-allele-classifiers).

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

## Acknowledgements

We would like to thank Anand Sastry, Jean-Christophe Lachance, Yara Seif, and Jason Hyun for helpful discussions and Marc Abrams for editing the manuscript. This research was supported by the NIAID grant (AI124316), the NIGMS (GM102098), and the Novo Nordisk Foundation Grant Number NNF10CC1016517.

## Author contributions

E.K. and B.O.P. conceived and designed the study. E.K. conducted all analysis, with contributions from L.Y., D.H., and J.M.M.. E.K., L.Y., D.H., J.M.M., and B.O.P. provided study oversight, wrote the manuscript, and edited the manuscript. B.O.P. managed the study. All authors reviewed and approved the final manuscript.

## Competing interests

The authors declare no competing interests.
