## [Peer Review File · Nature Communications]

Reviewers' Comments:

Reviewer #1:

Remarks to the Author:

The "A biochemically-interpretable machine learning classifier for microbial GWAS" manuscript by Erol, et al., addresses an important issue in the field of antibiotic resistance, namely a novel methodology to make mechanistic predictions of the role of antibiotic-resistance mutations in metabolism. The authors instead of using a classical correlative approach (i.e. GWAS), used a constraint-based modeling approach to predict flux changes in resistant vs sensitive TB strains. I find the methodology innovative and I can envisage many researchers in the field of systems biology and drug resistance (beyond antimicrobials) to adopt it. Although the topic can be of broad interest to the scientific community, in the present form, the manuscript is difficult to access. The description of the methodology is difficult to follow, such that I'm left wondering how robust predictions are and what are the really novel predictions and those which are used as "validation".

Major comments

From the introduction it is very difficult to understand what the authors are really trying to do and what are the key advantages of their procedure compared to classical GWAS. It might seem that a major goal of the method is the classification of AMR phenotypes or the identification of "key genetic determinants". However, because one key input to the model are a preselection of alleles previously found to relate to resistance, it is not surprising that the model is able to classify and identify AMR related mutations (see also following comments).

Only after reading half of the manuscript it becomes clear that one key advantage of the proposed methodology is to predict how such mutations would affect overall metabolism in resistant strains, leading to experimentally testable hypothesis of the role of flux changes in conferring or compensating for resistance.

Technically I'm concerned by the vastity of the space of possible allele-constraints and objective functions to be searched, and hence the robustness of model predictions. Somehow, I cannot find convincing evidence that the sampling is adequate enough to generate robust predictions.

Moreover, in Fig. 2b the authors showed that only very few MACs can generate adequate predictions differentiating resistant from sensitive strains. While this is presented as a positive aspect, I wonder whether these models could have emerged by chance from the sampling. For example, what would the results look like if the G matrix was randomized (e.g. random association between alleles and fluxes)? How would results in Fig. 2d and e would look like if only a subset of alleles were selected for training the model? How does AUCs compared to simply estimating the genetic distance between strains?

The interpretation of LOR-flux correlations is not clear. My expectation is that most flux rearrangements predicted by the model, which are not catalyzed by mutated enzymes, are indirect adaptive changes to the mutations. Hence, these changes are likely to be not directly involved in resistance mechanisms but rather in their compensation. The authors should clarify this point.

Moreover, I think the potential ability of the model to predict compensatory metabolic mechanisms could be a major selling point. If this is indeed one of the predictive value of such modeling approach, why the authors decided to focus only on drugs that act on metabolic enzymes? If the model can be applied also to drugs with non metabolic targets (e.g. ribosome or DNA replication inhibitors), the predictions would be entirely novel opening new opportunities for understand the role of metabolism in compensating antibiotic resistance.

It is not clear why the authors focused on 3 drugs, and what predictions are novel from those that look like more as a "sanity check" (e.g. katG). While I understand that experimental validation in AMR TB strains is perhaps out of the scope of this study, after reading the 3 case studies I'm left wondering what the model predictions really tell us, besides that there are changes in fluxes. The authors suggest that such changes should hint at selective pressures acting on the catalyzing enzymes. Are there any evidence supporting this? For example gene expression data showing that expression of these enzymes is significantly altered in resistant TB strains?. I actually suspect that most of the changes are indirect. The key question for me is whether these changes are simply fulfilling mass balance constraints, or if they could be driven by the selected objective function. The authors seem to completely neglect that MAC model also makes prediction of new objective

functions. Is there a specific reason for that? Could one use prediction of objective functions to better understand the role of metabolic changes in resistant strains?

The authors claim that their modeling approach (MAC) outperforms classic GWAS in predicting AMR. However it is not clear to me what are the evidence for that. The new methodology proposed by the authors is based on a preselection of alleles previously identified to be implicated to AMR, presumable by statistical analysis similar to GWAS. Hence, selected genes are already discriminative of sensitive vs resistant TB strains. The ranking argument (lines 408-417) seems an unfair comparison. Similarly to the following argument on the enrichment analysis. It is a triviality that MAC models emphasize the selection of pathway related to mutated enzymes involved in resistance, as most likely flux constraints on the AMR-related reactions will cause flux changes in the entire pathway.

Reviewer #2:

Remarks to the Author:

The research group led by Dr Pálsson has previously published genome-scale metabolic models for several bacterial pathogens, including *Salmonella* and *Staphylococcus aureus*; as well as constraint-based modelling methods to understand genotype-phenotype relationships using these genome-scale metabolic models. In this new study, the group aim to combine the power of metabolic modelling with a GWAS approach.

The authors note that genetic variants associated with drug resistance in *M. tuberculosis* often map to the same metabolic network, that is, they reflect adaptations in the same biochemical process in response to antibiotic selective pressures.

They developed method named Metabolic Allele Classifier (MAC) that takes the genome sequence of a particular TB strain and classifies it as either resistant or susceptible to a specific antibiotic. The authors used an existing dataset of whole-genome sequenced TB strains they had previously used.

They propose to incorporate metabolic network information as part of machine learning classifiers to facilitate the biological interpretation of microbial genome-wide association studies (GWAS). This is to me, the key and most innovative development of this work which, in my opinion, deserves being published in *Nature Communications*.

However, in its current form, the manuscript will not accessible to a wide audience due to abundance of technical terms throughout the text, which should normally be restricted to the Methods section. To facilitate the reading, the authors should put more emphasis on the biological interpretation of model parameters across all modelling steps.

Specific comments:

Methods

- The sentence in lines 503-505 is repeated again in lines 514-516.

- Line 615. Conventional GWAS and pathway analysis of allelic variants. The authors apply a GWAS to identify alleles significantly associated with AMR phenotypes applying an ANOVA F-test. The authors should apply the state-of-the-art GWAS model based on linear-mixed models that adjust for population structure as implemented in:

- Lees JA, Galardini M, Bentley SD, Weiser JN, Corander J. 2018. pyseer: a comprehensive tool for microbial pangenome-wide association studies ed. O. Stegle. *Bioinformatics* 34: 4310-4312.

Or

- Earle SG, Wu CH, Charlesworth J, Stoesser N, Gordon NC, Walker TM, Spencer CCA, Iqbal Z,

Clifton DA, Hopkins KL, et al. 2016. Identifying lineage effects when controlling for population structure improves power in bacterial association studies. *Nature Microbiology* 1: 16041.

Introduction

The authors need to introduce the term and concept of “metabolic flux” and how it relates to more familiar terms like “metabolic pathway”, “metabolic reaction”, “enzymes” or “protein-coding sequence (CDS)”.

The authors should also explain the biochemical rationale for proposing metabolic flux as the unit of association (from which significantly associated pathways and genetic loci and alleles are later derived) as opposed to using entire metabolic pathways or metabolic submodules as their preferred unit of association, as done in classical/conventional pathway-based GWAS analysis. I am not suggesting the latter is more valid, but instead more commonly seen in the GWAS literature.

Results:

Line 88. By using the “unique amino acid sequence” of proteins as “alleles” the authors restrict their analyses to genetic variants (SNPs and indels) that lead to non-synonymous amino acid changes (and frame-shift mutations?), that is, protein-altering variants. This is a valid approach, but the authors need to be more explicit about this and the fact they filter out synonymous amino acid changes and intergenic genetic variants. Specifically because they use the term “genetic variant matrix” which may lead the reader to think about a matrix of nucleotide alleles.

Line 89. “The corresponding AMR”, use “The corresponding drug susceptibility status for a strain is described...”

Line 90. Include what percentage of genes (in brackets) in the H37Rv genome these 1,011 correspond to.

Line 94. The authors state that the iEK1011 GEM includes 1,011 genes, but right after mention that 981 genes are found in the genomic dataset. How can the authors explain that 30 genes in the iEK1011 are not found any of the strains sequenced in their collection?

Lines 98 – 102. The authors need to comment more on and be more explicit about the antibiotic resistances they cannot model, that is, fluoroquinolones (DNA replication), rifampicin (RNA synthesis) and aminoglycosides (protein synthesis).

Related to this, sheet 1 in Supplementary Data File 1 does not seem to be complete. The ‘Paper’, ‘Mechanism of Action or Metabolic Effect’ and ‘Antibiotics’ columns do not contain text for all genes/rows. In Sheet 1, the column ‘Mutations’ has also a lot of empty cells.

Figure 1 Footnote. In text “GWAS data describing TB genome sequences”, avoid using the term ‘GWAS’ when referring to the genomic collection used, as GWAS analyses has not yet been applied in step a.

Line 129. The authors may want to cite one of their own articles (Orth JD, Thiele I, Palsson BØ. 2010. What is flux balance analysis? *Nature Biotechnology* 28: 245–248.) to introduce the reader to flux balance analysis.

Lines 166 – 166. I find difficult to interpret the relationship between flux states and alleles biologically, that is, how alleles (representing SNPs and indels in enzyme-coding genes) impose constraints on metabolic fluxes. In the review cited above (Orth et al. 2010), the authors explain that constraints can be used to represent genetic manipulations (such as gene knock-outs) by limiting metabolic reactions to zero flux.

Lines 171 - 180. Related to the point above, the authors need to include a better explanation on how “antibiotic-specific objective coefficients”, obtained after optimising the objective function from the data, can be interpreted biologically. How should the expressions “level of activity of metabolic pathways” and “fluxes activated by alleles” be interpreted?

Lines 196-197. The authors limited the set of alleles modelled by the MAC to those in AMR genes only. Does this reflect a limitation of the proposed metabolic modelling approach in the number of alleles that can be modelled at the same time? This is included in the Discussion but it will be helpful to justify this choice here.

Could this model be trained with all alleles in the genome (i.e. all CDS in iEK1011) to identify genes and metabolic processes not yet known to be involved in drug susceptibility?

Line 203-204. How can the number of high-quality MACs per antibiotic be interpreted? Is this a function of available sample sizes, that is, total number of susceptible and resistant strains tested for a particular antibiotic? Or is this a consequence of the number/complexity of metabolic processes governing susceptibility to a particular antibiotic? In other words, do pyrazinamide result in a higher number of high-quality MACs than cycloserine because the authors used a higher number of strains tested for pyrazinamide than to cycloserine? Or because resistance to pyrazinamide can result from multiple metabolic adaptations?

Line 233. Given that rifampicin resistance genes *rpoB* and *rpoC* are not in the GEM model, how can the authors interpret the best MACs for rifampicin?

Line 247. The authors need to be more explicit on why they focus on pyrazinamide, para-aminosalicylic acid and isoniazid; and the rationale for excluding the rest. It is understandable not to include the antibiotics they cannot model, that is, fluoroquinolones (DNA replication), rifampicin (RNA synthesis) and aminoglycosides (protein synthesis); but what about the rest?

Lines 258. Alleles in supplementary tables (in the tabs ending with `_MNC_allele_params`) should also be expressed as mutations (SNPs or indels) with respect to the H37Rv reference genome, using HGVS nomenclature (<https://www.hgvs.org/mutnomen/recs.html>). Also add a new column with the Rv locus name of each gene and metabolic pathways the gene belongs to (extracted from Supplementary File 2). This way readers will be able to relate drug resistance mutations and gene names they may be more familiar with to their metabolic pathway(s).

Lines 260-262. Indicate how many CDS in the H37Rv reference genome are included in the curated gene-pathway annotation, both as a number and percentage.

Figure 3b and similar. Indicate what mutation(s) each allele in the x-axis corresponds to.

Line 311. Do the authors mean *alr* by “alar”?

Lines 311 – 312. The authors identify 8 genes through the flux GWAS for para-aminosalicylic acid. At least four of these genes – *katG* (isoniazid), *inhA* (isoniazid), *pncA* (pyrazinamide) and *ald* (cycloserine) – are known to be involved in resistance to other drugs. This is not a limitation of their approach but the fact that clinical strains of *Mtb* that are resistant to last-line drugs (like para-aminosalicylic acid) are commonly resistant to other drugs too, that is, resistances commonly co-occur. In this regard, it does not make much sense to include *pncA* and *alr* alleles in Figure 4b. Thus, and as an example, the decreasing selection pressure in *pncA* identified here for PAS is most likely the result of pyrazinamide resistant strains in the para-aminosalicylic acid training set ($n=375$). The authors need to look at the co-occurrence and correlation of drug resistances in their training sets as this would help them interpret the GWAS results.

Line 352. The authors should indicate how many strains are resistant and sensitive in each training

dataset here, and anywhere else describing the size of training sets.

Line 355. The authors identified many more significant fluxes for isoniazid than for the other two described drugs. How can this be explained? Does the isoniazid training set contain a higher proportion of resistant strains than other drug training sets? Or does isoniazid resistance result from more diverse metabolic adaptations?

Lines 406 – 417. At the moment, the comparison with the classical GWAS results is rather unfair. The authors should use state-of-the-art GWAS methods that implement linear-mixed models.

Reviewer #3:

Remarks to the Author:

The manuscript by Palsson team presents a study of a novel methodology aiming to integrate mutation data into metabolic networks to provide interpretation to a so called “black box” machine learning models. As a case study authors use data from antibiotics resistance study that genotyped >1500 TB strains from previously published study.

Briefly, the authors perform a version of flux variability analysis (FVA) on the network that is constrained by mutations. Obtained flux boundaries from FVA then are then mapped to principal component space. Logistic regression with L1 regularisation then used on PCA-transformed variables to separate resistant from non-resistant strains. While authors written a nice biological story, the study has significant conceptual and technical pitfalls outlined in the following:

Major concerns:

Authors do not use any control for overfitting instead select models purely based on BIC criterion which just evaluates model “quality” as a function of number of parameters, it does not tell anything about generalisation of model, i.e. prediction performance on test set, from what I read, I believe the presented results are just fit to the data. Authors presented results on a held out set, that is in the majority of tested cases 2-3 larger than training dataset (Figure 2), which is very suspicious and probably technical error. However, if that is still the case the reason this could occur, is just simply because few common mutations makes TB resistance and one does not need to do FBA to explain them. It seems authors confuse “predictive” modeling with statistical inference, providing no QC analysis of regression model and call everything machine learning. It should be very clearly stated what is used for training what is for testing, how regularisation was tuned, on which data? Apart of multiple buzzwords I found manuscript very confusing to read. Due to this technical issue, all biological interpretation is questionable.

The motivation of manuscript is written using a very bold language, emphasising that machine learning is a “black-box”. While it is generally true for complex neural networks, in the present study authors use the most basic statistical logistic regression model that is extremely easy to interpret. The problem is the interpretation of complex multilayer networks with thousands of parameters, not the basic sigmoid function. Talking about interpretation, metabolic networks are low rank networks, e.g. reactions in linear pathways are highly correlative, making PCA on them lumps all fluxes into fewer components. E.g all fluxes will be correlated to glucose input and load to the same component, I don't understand how it simplifies the interpretation.

What I understood by reading few times the Methods, the formulation of MAC provided in is not what it is, objective function operates on PCA space space, the v in objective function is not v from FBA, is a linear combination of FVA v loaded on component. Is confusing to see the results of logistic regression as a within FBA framework which is traditionally formulated as LP problem.

Although, solving binary cross entropy with logistic function is a convex problem, with all the FVA flux discretisation, random allele sampling (why is it needed?) etc, I don't think can be formulated as a standard LP problem, which confuses the method presentation as integration of ML and FBA.

Minor issues:

Provided code does not work and naming conventions are not the same as in manuscript.

Specifically: Fails with example arguments, there is a bug involving argument parsing, maybe that's not reviewers job to fix it :)

Other issues:

Should provide sanity check tests (aka "self-tests"). Code looks messy (especially in the sense of structural integrity) so it makes me suspicious of its correctness.

Incomplete installation instructions: should mention the requirements.txt file

Incomplete execution instructions (MNC_DIR =?).

Refer to it at Metabolic Network Classifiers, not Allele Classifiers

We thank the reviewers for their valuable and constructive feedback. In light of the comments made by reviewers 1 and 2, we have changed our manuscript to hopefully be more accessible. To do this, we moved sections "Computing the Metabolic Allele Classifiers" and "Metabolic Allele Classifiers accurately and efficiently predict AMR" to the Supplementary and replaced them in the main text with two paragraphs in a section titled "Validation of Metabolic Allele Classifiers". Furthermore, we edited Figures 1 and 2 to highlight the biological interpretation of the results and to clarify MAC validation, respectively. We believe this makes the paper much more accessible by removing the complex details underlying the estimation of MAC parameters.

Reviewers' comments:

Reviewer #1 (Remarks to the Author):

The "A biochemically-interpretable machine learning classifier for microbial GWAS" manuscript by Erol, et al., addresses an important issue in the field of antibiotic resistance, namely a novel methodology to make mechanistic predictions of the role of antibiotic-resistance mutations in metabolism. The authors instead of using a classical correlative approach (i.e. GWAS), used a constraint-based modeling approach to predict flux changes in resistant vs sensitive TB strains.

I find the methodology innovative and I can envisage many researchers in the field of systems biology and drug resistance (beyond antimicrobials) to adopt it. Although the topic can be of broad interest to the scientific community, in the present form, the manuscript is difficult to access. The description of the methodology is difficult to follow, such that I'm left wondering how robust predictions are and what are the really novel predictions and those which are used as "validation".

Major comments

From the introduction it is very difficult to understand what the authors are really trying to do and what are the key advantages of their procedure compared to classical GWAS. It might seem that a major goal of the method is the classification of AMR phenotypes or the identification of "key genetic determinants". However, because one key input to the model are a preselection of alleles previously found to relate to resistance, it is not surprising that the model is able to classify and identify AMR related mutations (see also following comments).

We have now added the following text to the first paragraph of the introduction in order to make the point of our study clearer,

...These studies show that identified genetic associations have corresponding network-level associations that are highly informative of AMR mechanisms. However, current GWAS results only provide predictions for what alleles are most important, not their biochemical effects. ~~Machine learning models that incorporate biochemical network structure may thus lead to an enhanced understanding of AMR mechanisms through network-level interpretability of predictive genotype-phenotype maps¹³⁻¹⁵.~~ Therefore, machine learning models that incorporate biochemical network structure may naturally extend GWAS results by estimating network-level biochemical effects of identified alleles, leading to an enhanced understanding of AMR¹³⁻¹⁵.

We agree that it is important for the readers of the article to understand the source of alleles that were included in our analysis. The list of alleles utilized consists of both known AMR genes and unknown genes. We have added text to the section "Validation of Metabolic Allele Classifiers" to clarify this,

Since the computational cost of estimating MACs scales poorly with the number of alleles utilized, we limited the set of alleles modeled by the MAC to 237, describing 107 genes consisting of both known and unknown relations to AMR (**Supplementary File 1**). The known AMR genes provide validation cases while the unknown genes enable novel insights.

Only after reading half of the manuscript it becomes clear that one key advantage of the proposed methodology is to predict how such mutations would affect overall metabolism in resistant strains,

leading to experimentally testable hypothesis of the role of flux changes in conferring or compensating for resistance.

That is indeed the primary advantage of the MAC. We have now modified Figure 1c to portray these metabolic predictions of allelic effects.

Figure 1. A metabolic systems approach for genetic associations. (a) In this study, data describing TB genome sequences and AMR data types are integrated with a metabolic model to learn a biochemically-interpretable classifier, named Metabolic Allele Classifier (MAC). The MAC parameters consist of allele-specific flux capacity constraints, a , and an antibiotic-specific metabolic objective, c , both of which are inferred from the data. (b) The optimal MAC describes strain-specific polytopes in flux space that separate into resistant (R) and susceptible (S) regions. The MAC objective function, $c^T v$, is identified as normal to the plane that best separates R and S. (c) The learned MAC provides biochemically-based hypothesis of AMR mechanisms and allele-specific effects through interpretation of c and v . The genome-scale flux state of a strain, v , consists of fluxes that are directly activated by alleles (allelic fluxes) and those that are flux-balance consequences of the allele-activated fluxes (compensatory fluxes). Abbreviations: S, susceptible; R, resistant; AMR, antimicrobial resistance.

Technically I'm concerned by the vastity of the space of possible allele-constraints and objective functions to be searched, and hence the robustness of model predictions. Somehow, I cannot find convincing evidence that the sampling is adequate enough to generate robust predictions. Moreover, in Fig. 2b the authors showed that only very few MACs can generate adequate predictions differentiating resistant from sensitive strains. While this is presented as a positive aspect, I wonder

whether these models could have emerged by chance from the sampling. For example, what would the results look like if the G matrix was randomized (e.g. random association between alleles and fluxes)? How would results in Fig. 2d and e would look like if only a subset of alleles were selected for training the model? How does AUCs compared to simply estimating the genetic distance between strains?

We have now reorganized the two sections detailing the estimation and validation of MACs in order to emphasize the assessment of MAC predictions. In particular, Figure 2 was edited by removing estimation details and adding a panel describing the distribution of objective function weights for each antibiotic, shown below,

Figure 2. Validation of Metabolic Allele Classifiers. (a) Receiver operator characteristic (ROC) curves for MAC AMR predictions. **(b)** Histogram of median absolute objective function coefficients for pyrazinamide, para-aminosalicylic acid, and isoniazid MACs. For each antibiotic, the reaction variable corresponding to the primary AMR gene is colored pink with the gene noted in parenthesis. Abbreviations: AUC, area under the curve.

The high AUC scores on the large test set show that the predictions of AMR class are generalizable and not overfit. The recapitulation of primary AMR genes in the MAC objective functions show that they are robust and encode valuable information in the structure of G (identification of primary AMR genes by the MAC would not appear if the G matrix was random). These results show that the sampling is sufficient to generate consistent and robust results.

The interpretation of LOR-flux correlations is not clear. My expectation is that most flux rearrangements predicted by the model, which are not catalyzed by mutated enzymes, are indirect adaptive changes to the mutations. Hence, these changes are likely to be not directly involved in resistance mechanisms but rather in their compensation. The authors should clarify this point. Moreover, I think the potential ability of the model to predict compensatory metabolic mechanisms

could be a major selling point. If this is indeed one of the predictive value of such modeling approach, why the authors decided to focus only on drugs that act on metabolic enzymes? If the model can be applied also to drugs with non metabolic targets (e.g. ribosome or DNA replication inhibitors), the predictions would be entirely novel opening new opportunities for understand the role of metabolism in compensating antibiotic resistance.

We have now changed the text to explain the LOR-flux correlation.

...We then set out to understand the genetic basis for the flux associations by ~~testing the alleles of each gene for a linear correlation between flux and log odds ratio~~ identifying loci in which the AMR association of each allele was correlated with their flux distribution (“LOR-flux correlation”) (see **Methods**). The idea here is that ~~for the selection pressure at a particular locus to be significant, we expect the flux effects of alleles at the locus to be proportional to their statistical association with AMR~~ resistant alleles have different metabolic effects than susceptible alleles for key genes. These allele-specific flux differences underlie the AMR classification accuracy of the MAC.

The MAC certainly predicts compensatory metabolic mechanisms. We have changed Figure 1 to emphasize the identification of compensatory fluxes. We focused on drugs targeting metabolic enzymes because the primary AMR alleles are modeled in the MAC. For non-metabolic targets such as rifampicin which targets the ribosome, the MAC does not have any reaction variables that describe *rpoB* or *rpoC* alleles. Therefore, the resulting MAC fluxes will have no relationship to the ribosome and can't be interpreted as compensations to resistant *rpoB/rpoC* alleles.

It is not clear why the authors focused on 3 drugs, and what predictions are novel from those that look like more as a “sanity check” (e.g. *katG*). While I understand that experimental validation in AMR TB strains is perhaps out of the scope of this study, after reading the 3 case studies I'm left wondering what the model predictions really tell us, besides that there are changes in fluxes. The authors suggest that such changes should hint at selective pressures acting on the catalyzing enzymes. Are there any evidence supporting this? For example gene expression data showing that expression of these enzymes is significantly altered in resistant TB strains?. I actually suspect that most of the changes are indirect. The key question for me is whether these changes are simply fulfilling mass balance constraints, or if they could be driven by the selected objective function. The authors seem to completely neglect that MAC model also makes prediction of new objective functions. Is there a specific reason for that? Could one use prediction of objective functions to better understand the role of metabolic changes in resistant strains?

The choice to focus on 3 drugs was due to content limitations. The MAC enables biological interpretation of model parameters which consequently leads to more analysis and results. We have added text to explicitly state the rationale of these 3 antibiotics,

Below, we focus our analysis on three case studies: pyrazinamide, para-aminosalicylic acid, and isoniazid AMR. ~~These three antibiotics were chosen due to having both characterized and uncharacterized mechanisms underlying their associated alleles, allowing for both test cases and novel insights for the MAC.~~

Evidence supporting selective pressure acting on catalyzing enzymes was provided in Supplementary Table 1. We have now added a reference to this table in the first introductory paragraph.

Since the objective functions are the crux of our study, we have now edited Figure 1 to emphasize their biological interpretation (see response to 2nd comment) and analyzed them in Figure 2b. The prediction of compensatory flux is now highlighted in Figure 2b.

The authors claim that their modeling approach (MAC) outperforms classic GWAS in predicting AMR. However it is not clear to me what are the evidence for that. The new methodology proposed by the authors is based on a preselection of alleles previously identified to be implicated to AMR, presumable by statistical analysis similar to GWAS. Hence, selected genes are already discriminative of sensitive vs resistant TB strains. The ranking argument (lines 408-417) seems an unfair comparison. Similarly to the following argument on the enrichment analysis. It is a triviality that MAC models emphasize the selection of pathway related to mutated enzymes involved in resistance, as most likely flux constraints on the AMR-related reactions will cause flux changes in the entire pathway.

We agree with this good point that the comparison in gene identification ability is not fair since the allele set used for comparing GWAS and MAC was limited. We have now removed this comparison from the study. While the reviewer is correct that the ability of the MAC model to significantly improve pathway identification over conventional approaches is due to flux constraints, and thus may appear trivial, this result emphasizes an advantage of the MAC and provides basic insight to how the MAC works and interpretation of the predictions made.

Reviewer #2 (Remarks to the Author):

The research group led by Dr Palsson has previously published genome-scale metabolic models for several bacterial pathogens, including Salmonella and Staphylococcus aureus; as well as constraint-based modelling methods to understand genotype-phenotype relationships using these genome-scale metabolic models. In this new study, the group aim to combine the power of metabolic modelling with a GWAS approach.

The authors note that genetic variants associated with drug resistance in M. tuberculosis often map to the same metabolic network, that is, they reflect adaptations in the same biochemical process in response to antibiotic selective pressures.

They developed method named Metabolic Allele Classifier (MAC) that takes the genome sequence of a particular TB strain and classifies it as either resistant or susceptible to a specific antibiotic. The authors used an existing dataset of whole-genome sequenced TB strains they had previously used.

They propose to incorporate metabolic network information as part of machine learning classifiers to facilitate the biological interpretation of microbial genome-wide association studies (GWAS). This is to me, the key and most innovative development of this work which, in my opinion, deserves being published in Nature Communications.

However, in its current form, the manuscript will not accessible to a wide audience due to abundance of technical terms throughout the text, which should normally be restricted to the Methods section. To facilitate the reading, the authors should put more emphasis on the biological interpretation of model parameters across all modelling steps.

We thank the reviewer for their insightful comments. We have now replaced sections detailing the estimation process with a single section titled "Validation of Metabolic Network Classifiers". The previous sections have been moved the Supplementary Material. Furthermore, we have edited Figure

1 to illustrate the biological interpretation of the model parameters across the different modelling steps. We believe this makes the manuscript much more accessible.

Specific comments:

Methods

- The sentence in lines 503-505 is repeated again in lines 514-516.

We have removed the redundant sentences in lines 514-516.

- Line 615. Conventional GWAS and pathway analysis of allelic variants. The authors apply a GWAS to identify alleles significantly associated with AMR phenotypes applying an ANOVA F-test. The authors should apply the state-of-the-art GWAS model based on linear-mixed models that adjust for population structure as implemented in:

- Lees JA, Galardini M, Bentley SD, Weiser JN, Corander J. 2018. pyseer: a comprehensive tool for microbial pangenome-wide association studies ed. O. Stegle. Bioinformatics 34: 4310–4312.

Or

- Earle SG, Wu CH, Charlesworth J, Stoesser N, Gordon NC, Walker TM, Spencer CCA, Iqbal Z, Clifton DA, Hopkins KL, et al. 2016. Identifying lineage effects when controlling for population structure improves power in bacterial association studies. Nature Microbiology 1: 16041.

Because of concerns brought up by Reviewer 1, we have removed the comparison in gene identification between GWAS and MAC.

Introduction

The authors need to introduce the term and concept of “metabolic flux” and how it relates to more familiar terms like “metabolic pathway”, “metabolic reaction”, “enzymes” or “protein-coding sequence (CDS)”.

We have now added a Glossary to the Supplement that defines these terms and referenced it in the introduction as follows,

By computing **metabolic flux states (see Glossary for definition of terms)** ~~optimal pathway use~~ consistent with imposed biological constraints,

The authors should also explain the biochemical rationale for proposing metabolic flux as the unit of association (from which significantly associated pathways and genetic loci and alleles are later derived) as opposed to using entire metabolic pathways or metabolic submodules as their preferred unit of association, as done in classical/conventional pathway-based GWAS analysis. I am not suggesting the latter is more valid, but instead more commonly seen in the GWAS literature.

We have now included a Supplementary Notes section titled “How alleles connect to flux constraints” and referenced it in the introduction.

Results:

Line 88. By using the “unique amino acid sequence” of proteins as “alleles” the authors restrict their analyses to genetic variants (SNPs and indels) that lead to non-synonymous amino acid changes (and

frame-shift mutations?), that is, protein-altering variants. This is a valid approach, but the authors need to be more explicit about this and the fact they filter out synonymous amino acid changes and intergenic genetic variants. Specifically because they use the term “genetic variant matrix” which may lead the reader to think about a matrix of nucleotide alleles.

We have now included a sentence that explicitly describes this,

The acquired genetic variant matrix (**G**) of the 1,595 strains describes 3,739 protein-coding genes and their 12,762 allelic variants, where each variant is defined as a unique amino acid sequence for the protein coding gene. Our analysis therefore does not account for synonymous amino acid changes and intergenic genetic variants.

Line 89. “The corresponding AMR”, use “The corresponding drug susceptibility status for a strain is described...”

We have made this change in the manuscript.

Line 90. Include what percentage of genes (in brackets) in the H37Rv genome these 1,011 correspond to.

We have made this change in the manuscript. The 1,011 genes make up 26% of total H37Rv genes.

Line 94. The authors state that the iEK1011 GEM includes 1,011 genes, but right after mention that 981 genes are found in the genomic dataset. How can the authors explain that 30 genes in the iEK1011 are not found any of the strains sequenced in their collection?

The genomic dataset is filtered, so they may appear in the strains but were taken out during QA/QC purposes.

Lines 98 – 102. The authors need to comment more on and be more explicit about the antibiotic resistances they cannot model, that is, fluoroquinolones (DNA replication), rifampicin (RNA synthesis) and aminoglycosides (protein synthesis).

We have added the following sentence to the manuscript explicitly stating what antibiotics cannot be modeled by our approach,

AMR genes not explicitly accounted for in iEK1011 were primarily related to DNA transcription (e.g., *rpoB*) and transcriptional regulation (e.g., *embR*). The antibiotics rifampicin, ofloxacin, and streptomycin do not have AMR genes accounted for in iEK1011 and are therefore out of scope of our study.

Related to this, sheet 1 in Supplementary Data File 1 does not seem to be complete. The ‘Paper’, ‘Mechanism of Action or Metabolic Effect’ and ‘Antibiotics’ columns do not contain text for all genes/rows. In Sheet 1, the column ‘Mutations’ has also a lot of empty cells.

We have fixed Supplementary Data File 1 by completing the missing entries in the “Antibiotics” column and removing the “Mechanism of Action or Metabolic Effect” and “Paper” columns.

Figure 1 Footnote. In text "GWAS data describing TB genome sequences", avoid using the term 'GWAS' when referring to the genomic collection used, as GWAS analyses has not yet been applied in step a.

We have removed the term "GWAS" from Figure 1 caption.

Line 129. The authors may want to cite one of their own articles (Orth JD, Thiele I, Palsson BØ. 2010. What is flux balance analysis? Nature Biotechnology 28: 245–248.) to introduce the reader to flux balance analysis.

We have added the recommended citation of Orth JD et al 2010.

Lines 166 – 166. I find difficult to interpret the relationship between flux states and alleles biologically, that is, how alleles (representing SNPs and indels in enzyme-coding genes) impose constraints on metabolic fluxes. In the review cited above (Orth et al. 2010), the authors explain that constraints can be used to represent genetic manipulations (such as gene knock-outs) by limiting metabolic reactions to zero flux.

We have now included a Supplementary Notes section titled "How alleles connect to flux constraints" and referenced this section at the noted paragraph to clarify the biological relationship between flux states and alleles.

Lines 171 - 180. Related to the point above, the authors need to include a better explanation on how "antibiotic-specific objective coefficients", obtained after optimising the objective function from the data, can be interpreted biologically. How should the expressions "level of activity of metabolic pathways" and "fluxes activated by alleles" be interpreted?

We have now edited Figure 1 to include a panel that portrays the attributes of the objective function and their biological interpretation (panel c). The term "Fluxes activated by alleles" is also explained in this figure.

Lines 196-197. The authors limited the set of alleles modelled by the MAC to those in AMR genes only. Does this reflect a limitation of the proposed metabolic modelling approach in the number of alleles that can be modelled at the same time? This is included in the Discussion but it will helpful to justify this choice here.

Could this model be trained with all alleles in the genome (i.e. all CDS in iEK1011) to identify genes and metabolic processes not yet known to be involved in drug susceptibility?

The list of alleles utilized consists of both known AMR genes and unknown genes. For example, *ansP2* and *cysK2* are not AMR genes but were included in the gene list and were proposed as novel AMR determinants with a biochemical basis in this study. We have added text to the section "Validation of Metabolic Allele Classifiers" to clarify this,

Since the computational cost of estimating MACs scales poorly with the number of alleles utilized, we limited the set of alleles modeled by the MAC to 237, describing 107 genes consisting of both known and unknown relations to AMR (**Supplementary File 1**). The known AMR genes provide validation cases while the unknown genes enable novel insights.

Furthermore, the model could indeed be trained with all alleles in the genome (all CDS in iEK1011), but is too computationally expensive with the current estimation methodology. Training with all alleles may identify more novel genetic candidates and metabolic processes.

Line 203-204. How can the number of high-quality MACs per antibiotic be interpreted? Is this a function of available sample sizes, that is, total number of susceptible and resistant strains tested for a particular antibiotic? Or is this a consequence of the number/complexity of metabolic processes governing susceptibility to a particular antibiotic? In other words, do pyrazinamide result in a higher number of high-quality MACs than cycloserine because the authors used a higher number of strains tested for pyrazinamide than to cycloserine? Or because resistance to pyrazinamide can result from multiple metabolic adaptations?

The information regarding high-quality MACs and BIC-based MAC assessment has been moved to the Supplementary Discussion section since these are non-biological details of the MAC estimation process. The number of high-quality MACs reflects the parameter space of the fitted MACs, which is a function of both the training size and model complexity.

Line 233. Given that rifampicin resistance genes *rpoB* and *rpoC* are not in the GEM model, how can the authors interpret the best MACs for rifampicin?

If we fit a classical ML model between alleles and rifampicin AMR, but remove *rpoB* and *rpoC* alleles, the model would still predict AMR with relatively good accuracy because the presence/absence of other alleles such as *KatG* alleles are still informative of rifampicin AMR (due to the treatment regimen) (see ROC curve figure and table of top feature weights below). Similarly, *katG*-related metabolic fluxes become the strongest predictors of AMR in the MAC.

SVM WITH rpoB / rpoC top weights		SVM WITHOUT rpoB / rpoC top weights	
Rv0667_4	1743.960131	Rv3795_9	1159.732011
Rv3795_9	1151.653018	Rv2043c_10	962.280580
Rv2043c_10	1031.943588	Rv3854c_9	652.710354
Rv0667_3	949.504034	Rv0682_2	609.567899
Rv3854c_9	680.646019	Rv0682_4	466.811259
Rv0668_6	575.290323	Rv1908c_2	416.656474
Rv0682_2	534.663336	Rv1908c_5	404.005780
Rv1908c_2	456.476187	Rv0006_5	341.773561
Rv1908c_5	389.301972	Rv0016c_3	338.935188
Rv0682_4	381.810042	Rv3795_11	323.910103

Table of top 10 weighted features (absolute SVM coefficients) in two linear SVM models of rifampicin AMR that differ in presence of *rpoB*/*rpoC* alleles in the training dataset. The alleles of genes *rpoB* and *rpoC* (Rv0667 and Rv0668, respectively) are bolded. The *katG* alleles are those with Rv1908 text and appear in both top lists.

Line 247. The authors need to be more explicit on why they focus on pyrazinamide, para-aminosalicylic acid and isoniazid; and the rationale for excluding the rest. It is understandable not to include the antibiotics they cannot model, that is, fluoroquinolones (DNA replication), rifampicin (RNA synthesis) and aminoglycosides (protein synthesis); but what about the rest?

We focused on 3 antibiotics for content limitations. We have added text to explicitly state the rationale of these 3 antibiotics,

Below, we focus our analysis on three case studies: pyrazinamide, para-aminosalicylic acid, and isoniazid AMR. These three antibiotics were chosen due to having both characterized and uncharacterized mechanisms underlying their associated alleles, allowing for both test cases and novel insights for the MAC.

Lines 258. Alleles in supplementary tables (in the tabs ending with _MNC_allele_params) should also be expressed as mutations (SNPs or indels) with respect to the H37Rv reference genome, using HGVS nomenclature (<https://www.hgvs.org/mutnomen/recs.html>). Also add a new column with the Rv locus name of each gene and metabolic pathways the gene belongs to (extracted from Supplementary File 2). This way readers will be able relate drug resistance mutations and gene names they may be more familiar with to their metabolic pathway(s).

We have now added columns to the MAC_allele_params tab for each drug describing the amino acid sequence (aa_seq), set of mutations (mut), Rv locus name (Rv_locus_name), and gene pathways (pathways) for each allele.

Lines 260-262. Indicate how many CDS in the H37Rv reference genome are included in the curated gene-pathway annotation, both as a number and percentage.

We have now included this detail at the noted line. It is 32% of all H37Rv protein coding genes (1254/3906).

Figure 3b and similar. Indicate what mutation(s) each allele in the x-axis corresponds to.

Since many of these alleles map to numerous mutations (i.e., ppsA_4 contains 6 SNPs), displaying them along the x-axis in the figure is challenging. We have now referred to the Supplementary Data Files in the figure so the reader can find the list of mutations per allele.

Line 311. Do the authors mean alr by "alar"?

We have corrected the typo.

Lines 311 – 312. The authors identify 8 genes through the flux GWAS for para-aminosalicylic acid. At least four of these genes – katG (isoniazid), inhA (isoniazid), pncA (pyrazinamide) and ald (cycloserine) – are known to be involved in resistance to other drugs. This is not a limitation of their approach but the fact that clinical strains of Mtb that are resistant to last-line drugs (like para-aminosalicylic acid) are commonly resistant to other drugs too, that is, resistances commonly co-occur. In this regard, it does not make much sense to include pncA and alr alleles in Figure 4b. Thus, and as an example, the decreasing selection pressure in pncA identified here for PAS is most likely the result of pyrazinamide resistant strains in the para-aminosalicylic acid training set (n=375).

The authors need to look at the co-occurrence and correlation of drug resistances in their training sets as this would help them interpret the GWAS results.

The reviewer is correct that these genes identified for pyrazinamide are known determinants of other drugs. Despite this, we included the genes because we did not want to bias the portrayal of MAC results. We have added a sentence to the discussion mentioning the occurrence of these genes.

The identification of *alr* and *pncA*—known determinants of cycloserine and pyrazinamide, respectively—reflect the co-resistance of these strains and are not known to have selective pressure in para-aminosalicylic acid treatment.

Line 352. The authors should indicate how many strains are resistant and sensitive in each training dataset here, and anywhere else describing the size of training sets.

We have added text indicating the # of resistant and susceptible strains in each training dataset where they are described.

Line 355. The authors identified many more significant fluxes for isoniazid than for the other two described drugs. How can this be explained? Does the isoniazid training set contain a higher proportion of resistant strains than other drug training sets? Or does isoniazid resistance result from more diverse metabolic adaptations?

The larger number of significant fluxes for isoniazid than the other antibiotics is due to both the larger proportion of resistant samples and the metabolic effect of isoniazid AMR genes. Specifically, *katG* alleles directly perturb oxygen and hydroxide metabolites (Catalase peroxidase reaction) which directly impact a large number of subsystems (TCA, oxidative phosphorylation). Other isoniazid AMR genes impact the mycolic acid biochemical pathways.

Lines 406 – 417. At the moment, the comparison with the classical GWAS results is rather unfair. The authors should use state-of-the-art GWAS methods that implement linear-mixed models.

The reviewer makes a good point that the comparison in gene identification ability is not conclusive since the allele set used for comparing GWAS and MAC was limited. Furthermore, the comparison in identified gene sets is misleading since the MAC is not meant to compete against GWAS methods but instead extend their insights. We have now removed the comparison and instead focused the section on a comparison between MAC and conventional pathway analysis. The conventional pathway analysis makes use of the alleles identified by the MAC.

Reviewer #3 (Remarks to the Author):

The manuscript by Palsson team presents a study of a novel methodology aiming to integrate mutation data into metabolic networks to provide interpretation to a so called “black box” machine learning models. As a case study authors use data from antibiotics resistance study that genotyped >1500 TB strains from previously published study.

Briefly, the authors perform a version of flux variability analysis (FVA) on the network that is constrained by mutations. Obtained flux boundaries from FVA then are then mapped to principal component space. Logistic regression with L1 regularisation then used on PCA-transformed variables

to separate resistant from non-resistant strains. While authors written a nice biological story, the study has significant conceptual and technical pitfalls outlined in the following:

We thank the reviewer for their comments regarding the concepts and technical aspects of our methodology. From the reviewer's brief summary and major concerns, it is clear to us that the paper in its current form is inaccessible and leads to a misunderstanding of our presented method. We have clarified the reviewer's concerns below with detailed answers and revised the manuscript to hopefully clear up any possible misunderstandings. Specifically, we have moved the parts regarding estimation of MACs to the methods section, leaving only a paragraph describing the validation of the model on the test sets. In addition, Figures 1 and 2 have been edited to clarify the new insights provided by the MAC as well as validation. Furthermore, we have added supplementary information that contrasts the support vector machine with the MAC in order to help understand the MAC. We hope that this will achieve two things: (1) show that the model was not overfit, and (2) emphasize that the MAC is a linear program that classifies strains by solving the linear program and that the L1-PCR FVA model was used to estimate the parameters of the MAC.

Major concerns:

Authors do not use any control for overfitting instead select models purely based on BIC criterion which just evaluates model "quality" as a function of number of parameters, it does not tell anything about generalisation of model, i.e. prediction performance on test set, from what I read, I believe the presented results are just fit to the data. Authors presented results on a held out set, that is in the majority of tested cases 2-3 larger than training dataset (Figure 2), which is very suspicious and probably technical error. However, if that is still the case the reason this could occur, is just simply because few common mutations makes TB resistance and one does not need to do FBA to explain them. It seems authors confuse "predictive" modeling with statistical inference, providing no QC analysis of regression model and call everything machine learning. It should be very clearly stated what is used for training what is for testing, how regularisation was tuned, on which data? Apart of multiple buzzwords I found manuscript very confusing to read. Due to this technical issue, all biological interpretation is questionable.

Both the high AUC scores, which were determined using test sets, as well as the recapitulation of primary AMR genes show that the MAC was not overfit. While the BIC was used to select MACs, it was the test set AUC scores (not BIC) that allowed us to evaluate the generalisation and performance of the MAC. Our use of principal component regression, L1-regularization, and BIC criteria in determining the MAC parameters ensured that the model was sparse and explains the generalizability/performance of our model. The unusual train-test split is due to the computational challenge of estimating MACs, which scales poorly with the number of strains and alleles. The use of a large test set provides a robust evaluation of model performance/generalization and therefore provides stronger evidence that the estimated MACs were not overfit.

We have now replaced the sections detailing the MAC estimation process with a section titled "Validation of Metabolic Network Classifiers". We have additionally reworked Figure 2 to emphasize MAC validation. Figure 2a now shows the training and test splits.

The motivation of manuscript is written using a very bold language, emphasising that machine learning is a "black-box". While it is generally true for complex neural networks, in the present study authors use the most basic statistical logistic regression model that is extremely easy to interpret. The problem is the interpretation of complex multilayer networks with thousands of parameters, not the basic sigmoid function. Talking about interpretation, metabolic networks are low rank networks, e.g.

reactions in linear pathways are highly correlative, making PCA on them lumps all fluxes into fewer components. E.g all fluxes will be correlated to glucose input and load to the same component, I don't understand how it simplifies the interpretation.

The reviewer makes a good point that many machine learning models are interpretable. In fact, our previous work leverages the interpretability of simple machine learning models to identify primary AMR genes. However, current "black-box" ML models do not enable meaningful interpretation beyond identifying which features are most important (i.e., allele x determines AMR). Here, mechanistic knowledge regarding metabolic gene function and their interacting functions is integrated with an ML model such that the learned parameters describe the biochemical effects of alleles, providing new insights to AMR. We have now clarified this advantage in both the introduction and in Figure 1.

What I understood by reading few times the Methods, the formulation of MAC provided in is not what it is, objective function operates on PCA space space, the v in objective function is not v from FBA, is a linear combination of FVA v loaded on component. Is confusing to see the results of logistic regression as a within FBA framework which is traditionally formulated as LP problem. Although, solving binary cross entropy with logistic function is a convex problem, with all the FVA flux discretisation, random allele sampling (why is it needed?) etc, I don't think can be formulated as a standard LP problem, which confuses the method presentation as integration of ML and FBA.

The MAC objective function operates on the flux space (v), not PCA space. First, the PCA components are transformed back to FVA fluxes (vmax, vmins), such that the $\text{LogReg}(\text{PCA}(\text{FVA fluxes})) \sim \text{LogReg}(\text{FVA})$. We then relate the FVA variables to flux variables by mapping $v_{\max} = v_{\text{forward}}$, $v_{\min} = v_{\text{reverse}}$, so that $\text{LogReg}(\text{FVA}) \sim \text{LogReg}(v)$. This $\text{LogReg}(v)$ becomes the MAC objective function. We have added the following text to the methods section titled "Estimation of MAC objective function" to clarify this,

...We expect that the MAC predicts increasing or decreasing resistance as we maximize or minimize the objective value. ~~Specifically, we first transform the PCR variables back to popFVA variables through multiplication of v. We then translate the popFVA variables to reaction variables by replacing v_{\max} and v_{\min} by v_{forward} and v_{reverse} , respectively. The MAC objective coefficients of these forward and reverse reaction variables then take on the regression coefficients of the corresponding popFVA features.~~ Since the MAC objective function operates on the flux space, a series of mathematical transformations were taken to go from the PCR popFVA model to the MAC objective function (i.e., $\text{LogReg}(\text{PCA}(\text{FVA fluxes})) \rightarrow \text{LogReg}(v)$). We start with the PCA decomposition of the popFVA fluxes,

$$\text{PCA}(\text{FVA fluxes}) \rightarrow \mathbf{U} = \mathbf{XZ},$$

Where \mathbf{X} is the popFVA fluxes (strains, popFVA features), and \mathbf{U} describes the PCA components (strains, PCA components) and \mathbf{Z} has shape (popFVA features, PCA components). Fitting logistic regression to predict AMR using the PCA components gives the following equation,

$$\text{LogReg}(\text{PCA}(\text{FVA fluxes})) \rightarrow Y = b_0 + b_1 u_1 + \dots + b_k u_k,$$

Where u describes the k PCA components and b describes the LogReg coefficients. From this, we transform back to FVA space using the following mapping,

$$\mathbf{X} = \mathbf{UZ}^T$$

Which leads to the new LogReg equation,

$$\text{LogReg}(\text{FVA fluxes}) \rightarrow Y = b_0 + c_1 x_1 + \dots + c_m x_m,$$

Where $c_m = \sum_{i=1}^k z_{m,i} * b_i$ for m popFVA variables and k PCA components. The c values become the coefficients in the MAC objective function by representing the V_{\max} popFVA variables as V_{forward} flux variables and V_{\min} popFVA features as V_{reverse} flux variables.

The use of logistic regression is ideal for estimating an FBA objective function because it is a linear model. To our knowledge, this is the first time an LP itself has been used as a classifier and therefore may appear strange. LPs have previously been used to solve for a classifier (e.g., LPBoost), but not solved to classify. The distinction is subtle and thus may appear confusing.

The reviewer makes a good point that the MAC formulation is confusing within the context of machine learning classifiers since written optimization problems are usually used to describe how the classifier will be solved, not the classifier itself. The MAC itself is an LP but the optimization problem used to estimate the MAC is not an LP. We have now provided a comparison of the MAC to the Support Vector Machine in the Supplementary Discussion to help explain this distinction.

For an SVM, the learned classifier has the following form,

$$H_{\text{SVM}} = \text{sign}(\mathbf{w}^T \mathbf{x}_k + b),$$

Where $H > 0$ is resistant, $H < 0$ is susceptible, and \mathbf{x} describes the allele presence/absence vector of a particular strain. The parameters \mathbf{w} and b are learned from the data through optimization [2].

For the MAC, the learned classifier has the following form,

$$H_{\text{MAC}} = \text{sign}(\max \mathbf{c}^T \mathbf{v} + b \text{ subject to } \mathbf{Sv} = \mathbf{0}, \mathbf{v}^{\text{lb}} \leq \mathbf{v} \leq \mathbf{v}^{\text{ub}}, \mathbf{x}^T \mathbf{a}^{\text{lb}} < \mathbf{v} < \mathbf{x}^T \mathbf{a}^{\text{ub}}),$$

Where $H > 0$ is resistant, $H < 0$ is susceptible, \mathbf{x} describes the allele presence/absence vector of a particular strain, \mathbf{v} describes the flux state of the strain, \mathbf{S} is the stoichiometric matrix, and \mathbf{a} is the mapping of alleles to lb and ub flux constraints. The parameters \mathbf{w} , b , \mathbf{a}^{lb} , and \mathbf{a}^{ub} are learned from the data through a detailed multi-step optimization process (see **Methods**).

Minor issues:

Provided code does not work and naming conventions are not the same as in manuscript. Specifically: Fails with example arguments, there is a bug involving argument parsing, maybe that's not reviewers job to fix it :)

We apologize that the code repository did not work and have now fixed the issue.

Other issues:

Should provide sanity check tests (aka "self-tests"). Code looks messy (especially in the sense of structural integrity) so it makes me suspicious of its correctness.

We have now provided a Test Run section to the GitHub repository.

Incomplete installation instructions: should mention the requirements.txt file

We have now included the requirements.txt file in the installation instructions.

Incomplete execution instructions (MNC_DIR =?).

We have completed the execution instructions.

Refer to it at Metabolic Network Classifiers, not Allele Classifiers

The repository has now been renamed to Metabolic Allele Classifiers to reflect the name used in our study.

Reviewers' Comments:

Reviewer #1:

Remarks to the Author:

The authors adequately addressed all my concerns.

Reviewer #2:

Remarks to the Author:

The authors have incorporated all suggested changes. The manuscript reads better and is more accessible in the current form. I have no further suggestions.

Reviewer #3:

None

Our responses are in blue.

REVIEWERS' COMMENTS:

Reviewer #1 (Remarks to the Author):

The authors adequately addressed all my concerns.

We thank Reviewer #1 for their feedback.

Reviewer #2 (Remarks to the Author):

The authors have incorporated all suggested changes. The manuscript reads better and is more accessible in the current form. I have no further suggestions.

We thank Reviewer #2 for their feedback.